# The role of internal transcribed spacer 2 secondary structures in classifying mycoparasitic *Ampelomyces*

**Rosa E. Prahl**[1]*, **Shahjahan Khan**[2], **Ravinesh C. Deo**[3]

**1** School of Sciences, University of Southern Queensland, Toowoomba, Queensland, Australia, **2** School of Sciences, Centre for Health Research, Centre for Applied Climate Sciences, University of Southern Queensland, Toowoomba, Queensland, Australia, **3** School of Sciences, University of Southern Queensland, Toowoomba, Queensland, Australia

* u1114537@umail.usq.edu.au

**Data Availability Statement:** All relevant data are within the manuscript and its Supporting Information files.

**Funding:** This research was supported by the financial support of an Australian Postgraduate

## Abstract

Many fungi require specific growth conditions before they can be identified. Direct environmental DNA sequencing is advantageous, although for some taxa, specific primers need to be used for successful amplification of molecular markers. The internal transcribed spacer region is the preferred DNA barcode for fungi. However, inter- and intra-specific distances in ITS sequences highly vary among some fungal groups; consequently, it is not a solely reliable tool for species delineation. *Ampelomyces*, mycoparasites of the fungal phytopathogen order *Erysiphales*, can have ITS genetic differences up to 15%; this may lead to misidentification with other closely related unknown fungi. Indeed, *Ampelomyces* were initially misidentified as other pycnidial mycoparasites, but subsequent research showed that they differ in pycnidia morphology and culture characteristics. We investigated whether the ITS2 nucleotide content and secondary structure was different between *Ampelomyces* ITS2 sequences and those unrelated to this genus. To this end, we retrieved all ITS sequences referred to as *Ampelomyces* from the GenBank database. This analysis revealed that fungal ITS environmental DNA sequences are still being deposited in the database under the name *Ampelomyces*, but they do not belong to this genus. We also detected variations in the conserved hybridization model of the ITS2 proximal 5.8S and 28S stem from two *Ampelomyces* strains. Moreover, we suggested for the first time that pseudogenes form in the ITS region of this mycoparasite. A phylogenetic analysis based on ITS2 sequences-structures grouped the environmental sequences of putative *Ampelomyces* into a different clade from the *Ampelomyces*-containing clades. Indeed, when conducting ITS2 analysis, resolution of genetic distances between *Ampelomyces* and those putative *Ampelomyces* improved. Each clade represented a distinct consensus ITS2 S2, which suggested that different pre-ribosomal RNA (pre-rRNA) processes occur across different lineages. This study recommends the use of ITS2 S2s as an important tool to analyse environmental sequencing and unveiling the underlying evolutionary processes.

Award granted to Rosa E. Prahl from the Graduate Research School of University of Southern Queensland, Australia.

**Competing interests:** No authors have competing interests.

## Introduction

Ribosomes are RNA-protein complexes that are responsible for translating messenger RNA (mRNA) into protein. The eukaryotic nuclear ribosomal DNA (nrDNA) cistron encodes the rRNAs for the small (18S rRNA) and large (5.8S and 28S rRNA) ribosomal subunits. The nrDNA region also contains two spacers. The 1 internal transcribed spacer, ITS1, is located between the 18S and the 5.8S genes and the 2 internal transcribed spacer, ITS2, resides between the 5.8S and the 28S genes [1,2]. In order to fulfill the demand of protein production for various cellular processes, the nrDNA cistrons are transcribed at a high rate by the enzyme polymerase I [3], which generates pre-rRNA. Subsequently, this pre-rRNA undergoes multiple posttranscriptional processing events as well as ribosome biogenesis, which generates functional rRNAs comprising 5.8S, 18S and 28S units. During processing both spacers are removed [1,2].

rRNA genes are highly similar within and between individuals of the same species. Nevertheless, rRNA sequence variations are sufficiently divergent to differentiate one species from another. Consequently, the ITS1, 5.8S and ITS2, named the internal transcribed spacer (ITS) region, have been widely used for molecular identification and phylogenetic studies [4]. Indeed, the nrDNA ITS region is the preferred barcode in fungi classification [5]. However, the identification of fungal DNA environmental samples is challenging. Several taxonomical and phylogenetic fungal groups require appropriate cultural conditions before identification although many fungi are unculturable [6]. Furthermore, for some fungal groups, specific primers are required for successful amplification of other appropriate molecular markers [5,7]. In accordance with the taxonomy and phylogeny of the fungal group under study, the use of only one molecular marker can result in poor species discrimination resolution [8]. For a barcode gap, the interspecific genetic variability needs to be higher than the intraspecific variability. In fungi, the nrDNA ITS region has been widely used as a DNA barcode [8] as it has enough genetic variability to the species level, but sometimes it does not resolve closely phylogenetic related species and some fungi require extra identifiers for species delimitation within a determined genus or family [9]. For example, if some taxa have low ITS interspecific variability other molecular markers need to be used to precisely report genetic diversity [9]. Conversely, intragenomic variation of the ITS can result in overestimate the intraspecific variability in some fungi [9–12]. Thus, relying in the use of only one molecular marker in some fungi can result in poor species discrimination resolution and other molecular barcodes may be required as the accuracy of the identification can be compromised. Therefore, it is crucial to have an optimum barcode gap.

In addition, the ITS1 and ITS2 are assumed to evolve neutrally. However, it has been probed that they are subjected to secondary structural constraints [13,14], which may have an impact on phylogenetic inferences. For instance, the phylogeny of the species from the genus *Ampelomyces* are primarily based on the ITS region, where some contrasting results have been obtained. Interesting, the utility of *Ampelomyces* ITS2 secondary structures (S2s) to study its phylogeny has not been investigated even when several works have demonstrated ITS2 S2s can provide with important features to improve phylogenetic analyses [13,15].

*Ampelomyces* spp. are mycoparasites of the Erysiphaceae fungi [16], which are devastating plant pathogens [17,18] that are responsible for powdery mildew (PM) diseases that afflict important agricultural and ornamental crops worldwide [19]. Despite environmental and health risks [20], agrochemicals are mainly used to control and prevent these PM infections [21]. The potential use of *Ampelomyces* strains as biocontrol agents against PM fungi have been tested in both field [22–24] and laboratory environments [25,26]. Two different strains [25,27] have been commercialized as biofungicides with some degree of effectiveness [22]. In

order to advance the development of potent pest control products based on *Ampelomyces* strains, it is important to fully characterize their intra- and inter-population genetic diversity over space and time. Some studies suggest that the *Ampelomyces* mycoparasites are complex organisms that lack host specificity [28,29]. Nevertheless, temporal isolation can lead to genetic variability within populations [23]. *Ampelomyces* strains that infect apple powdery mildew (APM, caused by *Podosphaera leucotricha*) were found to be cryptic species of non-apple powdery mildew (non-APM) that resulted from an effective allopatric process. Furthermore, recombination events between APM and non-APM *Ampelomyces* were also detected, although their sexual reproductive stage was not observed [23].

Undoubtedly, population genetic studies in *Ampelomyces* are difficult to carry out. For example, some nucleotide sequences deposited in public databases under the name of *Ampelomyces* are not related to the mycoparasites by comparing their phylogeny and growth rates in culture [30,31]. In addition, most phylogenetic studies have been based on the ITS region [28–30,32–34] and other molecular markers may be required for species identification due to the high genetic divergence of ITS sequences from some *Ampelomyces* lineages that can reach up to 15% [24,30]. Nevertheless, the successful design of primers for other markers are difficult to obtain, which often occurs for other fungi. In order to resolve these issues and contribute to knowledge and future research in this area, we have investigated whether ITS2 S2s contain specific features that may be incorporated into phylogenetic analysis and provide with novel important information regarding evolutive processes occurring in *Ampelomyces*. Indeed, the utility of the ITS2 S2 to study the phylogeny of several organisms have been demonstrated [15,35] but it has not been investigated in *Ampelomyces*.

This study also has investigated whether the nucleotide sequences of the ITS region from fungi sampled in PM-free environments are still being deposited in the GenBank database as species of the genus *Ampelomyces*. Here, we also evaluated the potential use of the ITS2 S2 as an additional identifier of *Ampelomyces* against the related fungus *Didymella glomerata* (previously known as *Phoma glomerata*) [36] and other putative *Ampelomyces* fungi isolated from non-PM hosts [37–40]. This analysis included all of the nucleotide sequences comprising the ITS region from *Ampelomyces* spp. *sensu stricto* deposited in GenBank until the May 2020 period.

Lastly, we propose that the predicted S2 of ITS2 from *Ampelomyces* spp. obtained from the ITS2 Database (ITS2-DB) web server [41–45] can be utilized as a new guideline for future molecular identification of ITS sequences extracted from environmental DNA samples.

## Materials and methods

### Sequence acquisition from the GenBank database

The nucleotide sequences of the nrDNA ITS region of all species identified with the name of *Ampelomyces* were searched in the GenBank database using the query: *Ampelomyces* (in plain text). Partial sequences of the ITS region were not included in this analysis. In Table 1 in **S1 File** shows the details of all the 376 nucleotide sequences used in this study, which were derived from 376 *Ampelomyces* spp. *sensu stricto*. The former sequences were compared to the following sequences: (1) 10 sequences derived from putative *Ampelomyces* species (Groups 1 and 2), (2) four sequences belonging to *Didymella* species and (3) one sequence from *Phoma* species formed the outgroup taxa (**Tables 2–4 in S1 File**).

Group 1 contains two sequences of fungi identified as *Ampelomyces humuli*, which was derived from plant tissues of *Picea abies* decayed root [46] and from *Zea mays* [40]. Group 1 also contains three sequences from fungi known to form a different phylogenetic group from the 'true' species of the genus *Ampelomyces* [30,31] and are referred to as *A. humuli* (GenBank

sequence identifier (GI) AF035779), *Ampelomyces quercinus* (GI: AF035778) and *Ampelomyces* sp. (GI: U82452). *Ampelomyces quercinus* is currently known as *Nothophoma quercina* [36], **Table 2 in S1 File**.

Group 2 has five nucleotide sequences belonging to three fungi identified as *A. humuli* isolated from human nasal mucus [39] and soil [37], as well as two fungi identified as *Ampelomyces* spp. derived from creosote-treated crosstie waste [38] and air samples [47], **Table 3 in S1 File**. Finally, the outgroup is formed by ITS sequences belonging to four *Didymella* species [31,48–50] and one *Phoma* species [51], **Table 4 in S1 File**.

These groups were established in order to differentiate ITS sequences extracted from PMs that were unrelated to the genus *Ampelomyces* by cultural growth and phylogeny [30,31] from those sequences extracted from plant material and PM; and no plant material and human mucus, which were defined as groups 1 and 2, respectively.

## Selection and extraction of complete nrDNA ITS sequences from *Ampelomyces* spp. *sensu stricto*

Complete nucleotide sequences of ITS regions belonging to the *Ampelomyces* mycoparasites were selected as 'true' if their original references indicated that isolates or strains were derived from PM environments and if their cultural growth characteristics were also indicated. Any other relevant information such as photographs of fruiting bodies (pycnidia) and the type of phylogenetic analysis were considered important for the preliminary selection of key sequences.

We did not include partial sequences of ITS regions even when these were extracted from *Ampelomyces* species infecting PM fungi that had a phylogenetic analysis that correctly identified the mycoparasites from the outgroup taxa. This approach was required to correctly model their ITS2 S2s and to visualize the hybridization model of its proximal stem. All complete ITS sequences were retrieved directly from GenBank in the FASTA format. Redundant or partial sequences that were deposited by different authors that corresponded to the same type of culture were not selected even if they belonged to the genus *Ampelomyces*. We confirmed that these sequences belonged to the *Ampelomyces* mycoparasites by manually checking their original references.

## Comparison of ITS sequence length and nucleotide content between species of the genus *Ampelomyces* and to those *Ampelomyces* identified from PM-free environments

In this report, a total of 376 ITS sequences that correspond to the 'true' *Ampelomyces* species (**Table 1 in S1 File**) were compared to those from fungal species identified with the generic name of *Ampelomyces* and comprised by (1) putative *Ampelomyces* groups 1 and 2 and (2) outgroup taxa, (**Tables 2–4 in S1 File, respectively**).

## Sequence statistics of the nrDNA ITS region and its individual components (ITS1, 5.8S and ITS2)

Statistics of the nucleotide pair bases (G/C) and (A/T) expressed as total percentages from the ITS region and its components (ITS1, 5.8S and ITS2) were calculated for all groups using the Sequence Manipulation Suite at https://www.bioinformatics.org/sms2/dna_stats.html [52]. In order to determine the statistical significance of the sequence lengths and nucleotide contents of the ITS regions among fungal groups, the Kruskal-Wallis rank sum test for multiple independent samples was performed with the web server https://astatsa.com/KruskalWallisTest/ [53]. In brief, the differences among groups were evaluated by the Dunn post-hoc pairwise multiple comparison test after confirming that at least one group was different using the

Kruskal-Wallis test. Values were statistically significant with a $p < 0.05$. The Dunn $p$-values were calculated without $p$-value adjustment.

Partial 18S and 28S sequences that flank the ITS region were deleted before the analysis. Nevertheless, some DNA environmental sequences do not contain the common sequence motifs at the borders of the ITS region (partial 18S and 28S sequences) despite being deposited as complete ITS region sequences; and it is difficult to delimit them. Thus, the total nucleotide length of the complete ITS region for each fungus was normalized to the maximum length (697 base pairs) expected from ITS region sequences belonging to ascomycetes fungi that were amplified using the primers ITS1f and ITS4 as previously reported [54]; and statistical significance was determine by the Kruskal-Wallis rank sum test as described above.

## Insertion-deletion polymorphic analysis of the ITS1 among *Ampelomyces* groups and between unrelated groups

We investigate the distribution of insertion-deletion polymorphisms in ITS1 sequences that could cause sequence length variations. Multiple sequence alignment (MSA) of *Ampelomyces* ITS sequences extracted from each genus of PM were aligned independently, while those from putative *Ampelomyces* Groups 1 and 2 together with the outgroup were conducted with the 'multiple alignment using fast Fourier transform' (MAFFT) web service tool available at https://mafft.cbrc.jp/alignment/server/ [55–57]. The following parameters were used: (1) a scoring matrix 200 PAM/k = 2 and (2) a gap opening penalty of 1.53. The plots and alignments were executed with a threshold score of 39 (E = 8.4e-11). The sequences were aligned using the program MAFFTWS v7 and through the iterative refinement method L-INS-i (accuracy orientated) that includes local pairwise alignment. Resulting multiple alignments were saved in the FASTA format. All ITS1 sequences were extracted from the ITS alignment with MEGA-X V10.1.8 software [58]. Nucleotide polymorphism analysis was conducted with the DnaSP 6.0 software [59,60].

## ITS2 structure prediction

To evaluate the utility of ITS2 S2s, the ITS2 sequences from each *Ampelomyces* spp. *sensu stricto* and those from the putative *Ampelomyces* groups were extracted from the alignment between the 5.8S and 28S gene proximal stem motifs using the new version of the web interface Internal Transcribed Spacer 2 Ribosomal RNA Database, ITS2-DB, at http://its2.bioapps. biozentrum.uni-wuerzburg.de/ [41]. The complementary hybridization of both regions was observed using the ITS2-DB together with its 'Annotate' tool, which functions based on the hidden Markov models (HMMs), [61]. In order to predict the folding of the ITS2 S2, we used an expected value for detection of significant hits below 0.001 (E-value < 0.001) and HMMs for fungal organisms with a minimum size of 150 nucleotides. Visualization of the ITS2 S2 was conducted using the program PseudoViewer v3.0 at http://pseudoviewer.inha.ac.kr/ [62]. We selected only those S2s that were obtained by direct folding and preferred over other modelled S2s.

We also obtained minimum free energy structures of the ITS2 using the RNAfold online tool, which were visualized using a force directed graph layout (forna) [63] via the web application from ViennaRNA Web Services [64].

## Multiple ITS2 sequence-structure alignment and phylogenetic tree of *Ampelomyces* strains

To determine *Ampelomyces* lineages based on its ITS2 S2s, a simultaneous multiple sequence alignment of ITS2 S2s was firstly estimated. We used the online tool LocARNA v4.8.3 at

http://rna.informatik.uni-freiburg.de [65] for multiple alignment of RNA molecules. The input data were extracted from the ITS2-DB and comprised 25 ITS2 sequences and structures, which were obtained from the following sources: (1) 21 *Ampelomyces* spp., (2) two putative *Ampelomyces* Group 1 extracted from *Golovinomyces cichoracearum* (GI: U82452) and *P. abies* decayed root (GI: DQ093657), (3) one Putative *Ampelomyces* Group 2 extracted from creosote-treated crosstie waste (GI:GQ241274) and (4) one outgroup member from *Phoma herbarum* (GI: JF810528), (**Table 5 in S1 File**). The selected parameters consisted of a global alignment in LocARNA-P (probabilistic) mode where the complete input ITS2 sequences were aligned. For the alignment scoring, the default values were used. Thus, the values for structure weight, insertion-deletion (indel) opening score and indel score were 200, -800 and -50, respectively. The match score for the alignment of two identical sequences was 50, while the mismatch score for the alignment of two different sequences was 0. The RIBOSUM matrix was used to score sequence match/mismatch. The parameters of RNA folding energy were used with a temperature of 37˚C. The energy parameter settings were described by the Turner model 2004 [66].

The resulting multiple ITS2 sequence-structure alignment was analysed by the molecular evolutionary genetics analysis (MEGA)-X v10.1.8 software in order to calculate a maximum likelihood tree based on the Kimura two-parameter DNA model of evolution [67]. This analysis was conducted with a gamma distribution for the evolutionary rate among sites. The branches of the inferred unrooted tree were assayed using the bootstrap analysis with 1 000 replicates. The phylogram of 21 *Ampelomyces* sequence structure pairs was visualized using FigTree v1.4.4 software [68].

A second phylogenetic tree based on ITS sequences was built to compare the distribution of *Ampelomyces* clades and not related fungi with those obtained using ITS2 S2s, but the dataset consisted of 21 ITS from *Ampelomyces* spp. *sensu stricto* extracted from seven powdery mildew genera and four from the following sources: (1) two putative *Ampelomyces* Group 1extracted from *Golovinomyces cichoracearum* (GI: U82452) and *P. abies* decayed root (GI: DQ093657), (2) one Putative *Ampelomyces* Group 2 extracted from creosote-treated crosstie waste (GI: GQ241274) and (3) one outgroup member from *Phoma herbarum* (GI: JF810528), (**Table 5.1 in S1 File**) were utilized as the input dataset to conduct a MSA via the MAFFT web server at https://mafft.cbrc.jp/alignment/server/ [57] under the previous conditions. Next, phylogenetic analysis was performed with MEGA-X software [58] using the maximum likelihood method and the Tamura-Nei model [69] with gamma distribution among the sites. The branches of the inferred unrooted tree were assayed using bootstrap analysis with 1 000 replicates. The phylogram was visualized using FigTree v1.4.4 software [68].

For a comprehensible analysis, a second phylogenetic tree was constructed as described above, but the input data consisted of 26 ITS2 S2s from *Ampelomyces* spp. *sensu stricto* extracted from seven powdery mildew genera and four from the following sources: (1) Group 1, putative *Ampelomyces* extracted from *P. abies* (GI: DQ093657); (2) Group 2 that consisted of creosote-treated crosstie waste (GI: GQ241274) and (3) two of the outgroup members, *D. glomerata* (GI: MH864401) from soil and *Malus sylvestris* (GI: JF810528), (**Table 5.2 in S1 File**).

## Evolutionary distance estimation among *Ampelomyces* lineages and putative *Ampelomyces*

The resolution power of the phylogeny based on the ITS2 S2s to distinguish the 'true' *Ampelomyces* from putative *Ampelomyces* was evaluated by comparing the genetic divergences obtained among groups based on the simultaneous ITS2 sequences and structures alignment,

and on the MSA of complete ITS sequences. The input data was the 25 ITS2 sequences consisted of 21 from *Ampelomyces* spp., three from putative *Ampelomyces* and one from the outgroup taxa as described in the previous section, **Table 5 and 5.1 in S1 File**.

The mean sequence divergence values among the major clades were estimated using MEGA-X software with the log-determinant (log-det) method and the Tamura-Kumar model [70]. The rate variation among sites was modelled using a gamma distribution (with the shape parameter = 5) and standard errors of the estimate distances were calculated with MEGA-X software using the bootstrap method.

### Prediction of consensus ITS2 S2s from *Ampelomyces*

For comparative purposes of sequence-structure motifs among fungal groups, the resulting simultaneous alignment of ITS2 sequences and structures obtained via the ITS2-DB was used as input into the 4SALE v1.7.1 software [71,72] to obtain consensus ITS2 S2s from the following sources: (1) five from *Ampelomyces* spp. extracted from *Arthrocladiella mougeotii*; (2) four from *P. leucotricha*; (3) one from *Podosphaera ferruginea*; (4) four from *Erysiphe necator*; (5) one from *Uncinula necator*; (6) four from *Podosphaera fusca* (GI: DQ490745, DQ490747, DQ490754 and DQ490757); (7) one from *Podosphaera xanthii* (GI: DQ490759) and (8) four from putative *Ampelomyces* (GI: DQ093657, GQ241274, JF810528 and U82452).

## Results

The first aim of this study was to verify the authenticity of the nucleotide sequences of the ITS regions that were deposited in GenBank under the name *Ampelomyces*. Until May 2020, 808 sequences were retrieved and 19% were derived from environmental sampling of fungi or plant tissues. Amongst all nucleotide sequences belonging to *Ampelomyces* spp. *sensu stricto*, 68% corresponded to the ITS region (complete and partial sequences), 27% were from the *actin1* gene, 2.97% corresponded to nucleotide sequences from the small subunit ribosomal (18S) and the large subunit ribosomal (28S) genes (complete and partial sequences) and 0.98% corresponded to those from microsatellites and the sequence of the complete genome of *Ampelomyces quisqualis* strain MHLAC05119 [8], (**Table 1 in S1 File**). Since most of the sequences corresponded to the nrDNA ITS region, we focused our analysis on complete sequences of this molecular marker that reduced the dataset to 376 sequences.

### *Ampelomyces* spp. ITS nucleotide content and sequence length do not resemble those extracted from environmental DNA and human mucus samples

From the dataset of *Ampelomyces* spp. *sensu stricto* extracted from seven genera of PM fungi (*Arthrocladiella*, *Erysiphe*, *Golovinomyces*, *Neoerysiphe*, *Oidium* sp. subgenus *Pseudoidium*, *Phyllactinia* and *Podosphaera*), we found that the sequence lengths of the complete ITS regions (492–502 bp) and their constituents, the ITS1 (182–193 bp), and ITS2 (149–155 bp) varied across the whole population, **Table 2 in S1 File**.

In contrast, the sequence length from the 5.8S ribosomal gene (157 bp) remained constant. The A/T and G/C content for both spacers and the 5.8S ribosomal gene were above and below 50%, respectively. The lowest G/C content value (39.25%) was observed for the ITS1 (**Table 1**), **Table 3 in S1 File**.

In order to verify if the complete fungal ITS sequences from environmental DNA are related to those from *Ampelomyces* lineages, we have investigated whether they can be differentiated by comparing nucleotide content and sequence length of their complete nrDNA ITS

**Table 1. Characterization of the ITS region and their constituents ITS1, 5.8S and ITS2 from *Ampelomyces* spp.**

|  | ITS | ITS1 | 5.8S | ITS2 |
|---|---|---|---|---|
| **Sequence length (bp)** | 492–502 | 182–193 | 157 | 149–155 |
| **Average length (S.E.M.)** | 498 (0.004) | 188 (0.005) | 157 (0.00) | 152 (0.002) |
| **A/T content (%) range** | 54.05–58.92 | 54.59–60.75 | 53.5–57.32 | 50.00–58.82 |
| **G/C content (%) range** | 41.08–45.95 | 39.25–46.5 | 42.68–46.5 | 41.18–50.00 |

n = 376 *Ampelomyces* ITS sequences. Abbreviations: Base pair (bp); Standard error of the mean (S.E.M.).

regions. In addition, we used five ITS sequences retrieved from species of the *Didymella* and *Phoma* genera, as a reference outgroup. These species are commonly used as outgroup taxa in phylogenetic studies of the *Ampelomyces* mycoparasites [23,33]. Thus, we compared the sequence length and nucleotide content of the ITS sequences from *Ampelomyces* spp. *sensu stricto* to ITS sequences belonging to two groups of putative *Ampelomyces*. Group 1 contains sequences from *Ampelomyces heraclei*, *A. humuli*, and *Ampelomyces* sp. that were previously shown to be unrelated to the genus *Ampelomyces* [30,31]. It also contains two other sequences of fungi classified as *A. humuli*, which were isolated from plant material (*P. abies* and *Zea mays*) (**Table 4 in S1 File**).

On the other hand, Group 2 contains sequences extracted from human and environmental DNA samples, which were classified under the generic name of *Ampelomyces* (**Table 5 in S1 File**). We found that the average ITS sequence length for Groups 1 (453.6 bp ± 0.92) and 2 (453 bp ± 1.84), **Table 2**, differed significantly ($p$-values: $< 0.001$) by 50 bp when compared to those extracted from the entire *Ampelomyces* population (498 bp ± 0.004). Indeed, the average sequence lengths of the putative *Ampelomyces* ITSs were similar to those extracted from species of the *Didymella* and *Phoma* genera (452 bp ±0.0), (**Table 6 in S1 File**).

Several fungal environmental DNA ITS samples from GenBank do not contain nucleotide sequence information at the borders flanking the complete ITS region. Thus, all of the sequence lengths were normalized to the maximum sequence length that can be obtained for fungal species ascomycetes using the primers ITS1f and ITS4 as previously described [54]. After normalizing the sequence lengths, we observed the same differences in length variation of ITS sequences among all fungal groups (**S2 File, Table 7 and 7.1 in S1 File**) as described above. In addition, the normalized sequence length of the complete ITS region from the *Ampelomyces* lineages is significantly ($p$-values: $< 0.001$) higher than those from the putative *Ampelomyces* (**Fig 1**), **S2 File, Table 7.1 in S1 File**.

On the other hand, the normalized sequence lengths of the 5.8S gene and the ITS2 were similar across all the fungal groups. Conversely, the normalized sequence lengths of the ITS1

**Table 2. Variations in ITS region sequence length can distinguish between *Ampelomyces* spp. and misidentified fungi.**

|  | ITS1 (bp) | 5.8S (bp) | ITS2 (bp) | ITS (bp) |
|---|---|---|---|---|
| ***Ampelomyces* spp. *sensu stricto*** | 182–193 | 157 | 149–155 | 492–502 |
| **Putative *Ampelomyces*, Group 1** | 139–143 | 156–157 | 156–157 | 452–457 |
| **Putative *Ampelomyces*, Group 2** | 140–142 | 157 | 148–157 | 446–456 |
| **Outgroup taxa** | 139 | 157 | 156 | 452 |

The sequence lengths of the ITS region from *Ampelomyces* spp. (n = 376) were significantly higher than those from putative *Ampelomyces* Groups 1 and 2 (n = 5 each) as well as the outgroup (n = 5) with a Kruskal-Wallis chi-squared statistic value of 53.27 and $p$-values of 1.598204e-11. No differences in the sequence lengths of the ITS regions were observed between the misidentified fungi and the outgroup taxa with $p$-values of 1.0 and 3 degrees of freedom.

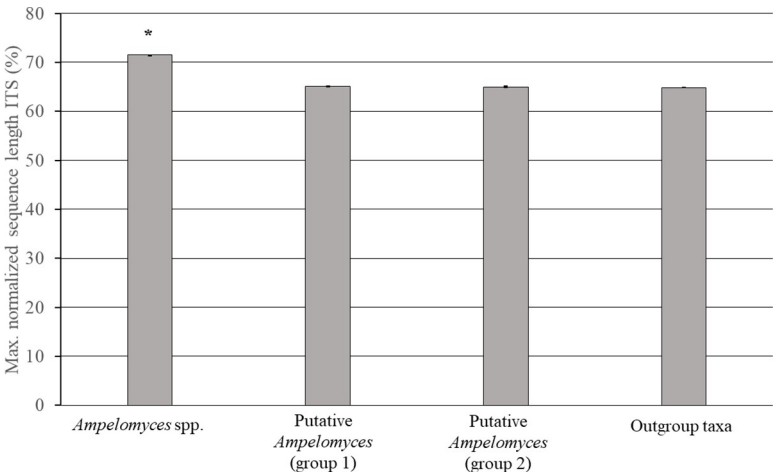

**Fig 1. Normalized nrDNA ITS sequence lengths from environmental fungi are similar to those from the outgroup.** The graph shows the normalized sequence length (%) of the ITS region from *Ampelomyces* strains (n = 376), putative *Ampelomyces* Groups 1 (n = 5) and 2 (n = 5) as well as the outgroup (n = 5). The normalized ITS sequence length of the *Ampelomyces* strains was significantly higher than those from putative *Ampelomyces* Groups 1 and 2 as well as from fungi belonging to the outgroup taxa, with a Kruskal-Wallis chi-squared statistic value of 53.27, a *p*-value of 1.59e-11 and 3 degrees of freedom. The error bars indicate the S.E.M. and the * indicates the statistical significance at the 5% confidence level.

from *Ampelomyces* spp. were significantly (*p*-values: < 0.001) higher than those from Groups 1 and 2 as well as the outgroup taxa (**Table 3**).

In addition, the A/T content of the ITS sequences from *Ampelomyces* spp. (**Fig 2A**) were significantly (*p*-values: < 0.001) higher than those from putative *Ampelomyces* Groups 1 and 2 as well as the outgroup taxa based on the Kruskal-Wallis test and Dunn's post-hoc test for multiple comparisons. Conversely, the C/G content (**Fig 2B**) around the median of the ITS sequences from *Ampelomyces* spp. were significantly (*p*-values: < 0.001) lower than those from putative *Ampelomyces* groups and the outgroup taxa based on the Kruskal-Wallis test and Dunn's post-hoc test for multiple comparisons.

Even though this study was not focused on identifying the ITS sequences from PM-free environments, we have found that the A/T and C/G median values of fungal Group 1 overlapped the median value from the outgroup. This indicated that the putative *Ampelomyces* from Group 1 are likely related to the outgroup taxa. We also observed that the nucleotide content values from fungal Group 2 were more varied, but not statistically different from fungal Group 1 (*p*-value: 0.98 and significance at 5%) and the outgroup taxa (*p*-value: 0.95 and

**Table 3. The maximum normalized sequence lengths of the 5.8S gene and ITS2 spacer were conserved among all fungal groups.**

|  | ITS1 (%) | 5.8S (%) | ITS2 (%) |
|---|---|---|---|
| ***Ampelomyces* spp. *sensu stricto*, n = 376** | 27.01 (0.00) | 22.52 (4.9.e-6) | 21.85 (0.00) |
| **Putative *Ampelomyces*, Group 1, n = 5** | 20.14 (0.04) | 22.46 (0.14) | 22.46 (0.01) |
| **Putative *Ampelomyces*, Group 2, n = 5** | 20.25 (0.02) | 22.52 (0.00) | 22.20 (0.09) |
| **Outgroup taxa, n = 5** | 19.94 (0.00) | 22.52 (0.00) | 22.38 (0.00) |

Only the normalized ITS1 sequence length from *Ampelomyces* spp. *sensu stricto* was significantly different to those from the other groups, with a Kruskal-Wallis chi-squared statistic value of 46.26 and a *p*-value of 4.972997e-10 with 3 degrees of freedom. The S.E.M. is indicated in parenthesis.

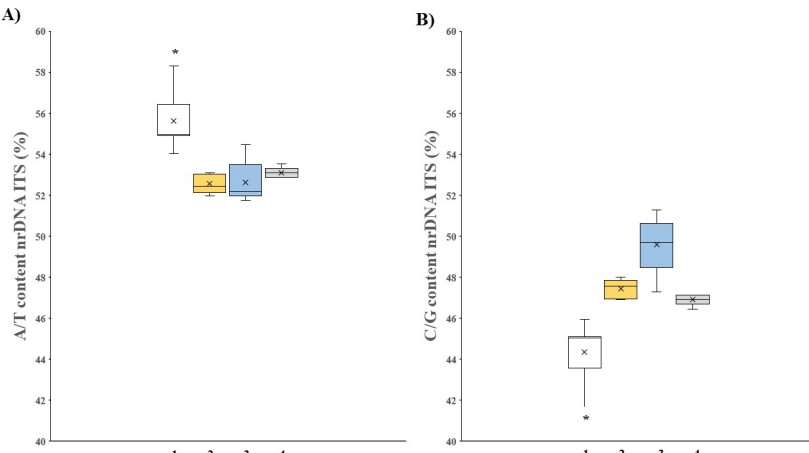

**Fig 2. Differences in ITS nucleotide contents help in distinguishing putative *Ampelomyces* from the 'true' *Ampelomyces*.** (A) The A/T and (B) C/G content values around the median for ITS sequences from *Ampelomyces* spp. (1; n = 376), putative *Ampelomyces* Groups 1 (2; n = 5) and 2 (3; n = 5) as well as the outgroup (4; n = 5). For each box plot, the central line as well as the top and bottom of each box represent the median, the third and first quartile, respectively. The whiskers indicate the maximum and minimum values, while circles above each box plot represent the outliers of each dataset. The statistical significance of the results was assessed using the Kruskal-Wallis test with Dunn's post-hoc test for multiple comparisons (*p*-values< 0.05) with 3 degrees of freedom. The differences were statistically significant at the 5% confidence level.

significance at 5%). In summary, no statistically significant differences in the A/T and C/G content were found between putative *Ampelomyces* and the outgroup taxa.

## Analysis of deletion/insertion polymorphisms in *Ampelomyces* ITS1 sequences and unrelated sequences

We found that the total number of indels (I = 8) between groups of *Ampelomyces* isolated from the same PM genera varied as assessed with DnaSP software [59,60]. Visualization of the MSA comprised of 376 ITS1 sequences, revealed that *Ampelomyces* ITS1 sequences extracted from PM of the genera *Erysiphe*, *Golovinomyces* and *Podosphaera* have indels that are evenly distributed along the sequence. In contrast, the ITS1s from *Ampelomyces* extracted from *Podosphaera fusca* in Korea, contained indels, which are more distributed at the beginning of the ITS1 sequence (first 100 nucleotides), while the same *Ampelomyces* sampled in China harboured indels at the end of the sequence. Conversely, *Ampelomyces* extracted from the genera *Arthrocladiella*, *Oidium* and the subgenus *Pseudoidium* had indels at the beginning and at the end of the sequence, while *Ampelomyces* derived from *Phyllactinia* contained indels localized at the end of the sequence. We also compared the number of indels (I = 8) for all *Ampelomyces* ITS1 sequences with those from putative *Ampelomyces* Groups 1 and 2 as well as the outgroup taxa (I = 6) where the ITS1 sequences for all outgroup samples contained indels at various positions across the sequence.

## Variations in the hybridization model of a proximal stem containing the 5.8S and 28S motifs were found in two *Ampelomyces* strains

We further characterized the ITS sequences by modelling the hybridization of a proximal stem containing the 5.8S and 28S motifs that delimit the S2 of the ITS2 (**Table 8 in S1 File**). Among the 376 ITS sequences from the 'true' *Ampelomyces* spp., 257 were not amenable for modelling with the ITS2-DB web server because these sequences contained a short 28S motif. Nevertheless, 120 ITS2 sequences were modelled and 118 exhibited the typical hybridization of

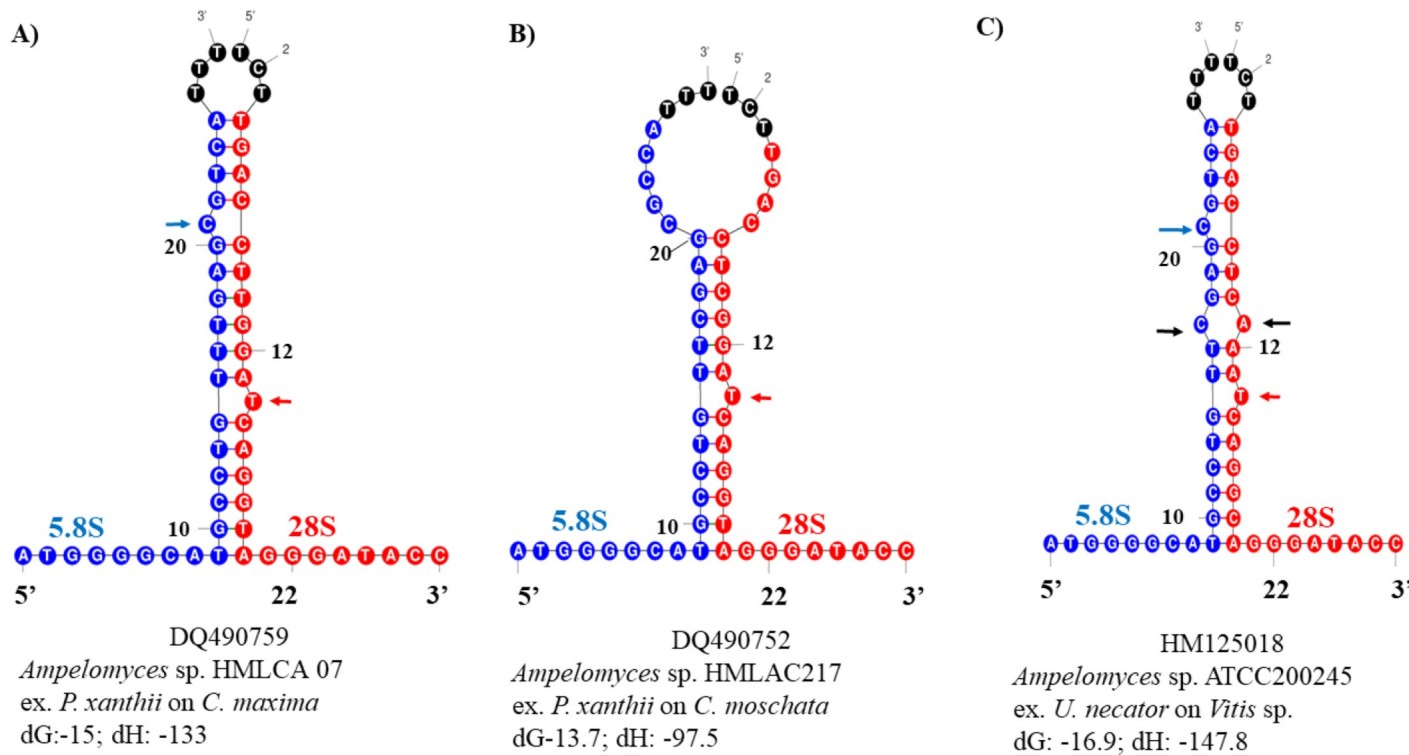

**Fig 3. Two variations in the hybridization model of the proximal stem region were found in *Ampelomyces*.** (A) The typical 5.8S and 28S hybridization model was obtained from an ITS2 sequence extracted from *Ampelomyces* conidia infecting *P. xanthii*. (B) A variation in the 5.8S and 28S hybridization model detected in *Ampelomyces* sp. infecting *P. xanthii*. (C) Another variation in the 5.8S and 28S hybridization model obtained from *Ampelomyces* sp. infecting *E. necator* (Section *Uncinula*) chasmothecia. The blue nucleotides denote the 5.8S strand, the red nucleotides indicate the 28S strand and the black nucleotides comprise the flanking regions of the ITS2 spacer. The formation of an internal loop is indicated by the two black arrows. The typical free nucleotides found in the 5.8S and 28S strands are indicated with blue and red arrows, respectively. The GenBank accession number is shown below each structure. The Gibbs free energy (dG) and enthalpy (dH) values are shown after the taxon name.

the proximal stem containing 5.8S and 28S motifs with a Gibbs free energy (dG) of -19 and enthalpy (dH) of -147.1 for the ensemble (**Table 8 in S1 File**). This S2 is formed by an imperfect stem that harbors one free nucleotide on the 5.8S strand and one free nucleotide on the 28S strand. Interestingly, variations in the dG and dH were detected from three other sequences. For example, DQ490759 had a dG and dH of -15 and -133, respectively, but the assembly of the typical proximal stem was not modified (**Fig 3A**).

Conversely, the DQ490752 and HM125018 sequences exhibited variations in the model along with thermodynamic changes in energies -13.7 and -16.9, respectively. DQ490752 (**Fig 3B**) corresponded to *Ampelomyces* sp. isolated from the conidia of *P. xanthii* infecting *Cucurbita*. The typical single free nucleotide in the 5.8S motif is absent and it is directly substituted with unaligned nucleotides from both strands that terminate in three free nucleotides. Another variation of the model was identified in the *Ampelomyces* ITS sequence extracted from the chasmothecia of *Erysiphe necator* (Section *Uncinula*) on grapevine bark (**Fig 3C**). In this model variant, the typical single free nucleotide from the 5.8S motif is absent and it is directly substituted with unaligned nucleotides from both strands that terminate in three free nucleotides. In addition, the dG and dH values for the ensemble of the typical hybridization model of the proximal stem from 118 *Ampelomyces* ITS2 sequences was -19 and -147.1, respectively. Even though we modelled a small sample of predicted hybridized stems, it appeared that stems with dGs less than -13.7 destabilized this secondary structure while values energy between -15 and -19 confer to the structure with some flexibility.

## One common ITS2 S2 was found across *Ampelomyces* spp. extracted from seven PM genera

From the dataset comprised of 120 ITS2 sequences, only 87 were directly folded and modelled (**Table 8 in S1 File**). We obtained seven different models. All contained a core structure with four helices (I–IV) and helix III was the longest (**Fig 4A**). All helices III contained a motif that varied across the *Ampelomyces* lineages (**Fig 4A and 4B**).

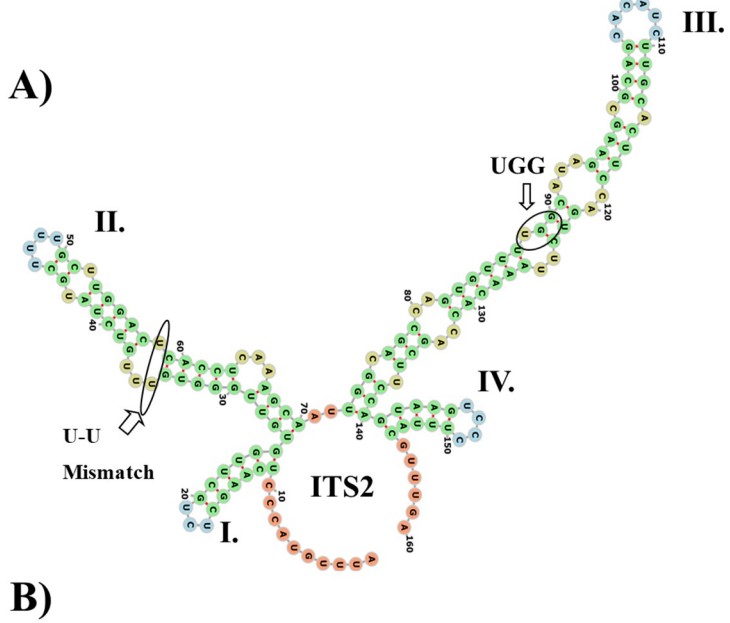

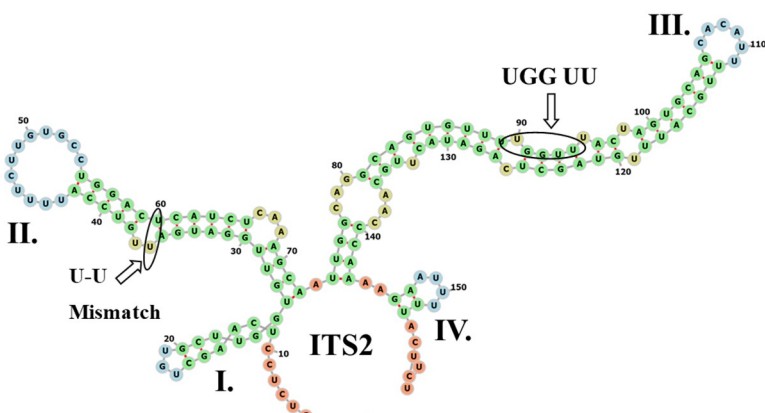

**Fig 4. Common features of the ITS2 S2 from *Ampelomyces*.** (A) The ITS2 S2 from *Ampelomyces* extracted from seven PM genera. The ITS2 contained a core structure with four helices (I–IV), a U-U mismatch on helix II and a large helix III containing a UGG motif. (B) This ITS2 S2 was found in *Ampelomyces* sampled in China (GI: DQ490752). The UGGU and UU motifs were located in helix III. The minimum free energy structures were calculated with RNAfold and visualized using a force directed graph layout with the Vienna RNA Web Services [64]. The helices are indicated with Roman numerals (I–IV). The U-U mismatch on helix II and the UGG, UGGU and UU motifs are indicated with arrows.

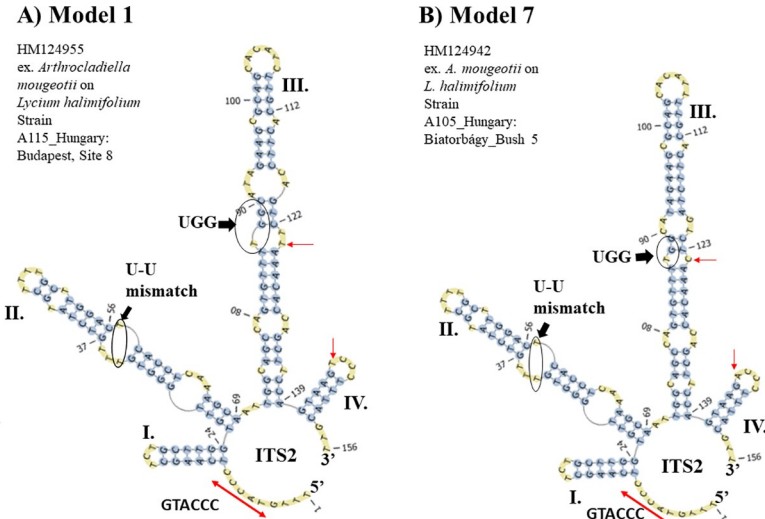

**Fig 5. The most common ITS2 S2 (Model 1) found in the *Ampelomyces* population.** (A) The ITS2 S2 Model 1 derived from *Ampelomyces* extracted from seven PM genera. (B) The ITS2 S2 Model 7 derived from *Ampelomyces* extracted from *A. mougeotii.* Compared to Model 1, Model 7 contains several nucleotide substitutions. Specifically, nucleotides cytosine and uracil in helices III and IV, respectively of Model 1 are present as uracil and thymine in the helices III and IV, respectively of Model 7; (red arrows indicate nucleotide substitutions). The helices are indicated by Roman numerals (I–IV). The following shared ITS2 features are shown in black arrows and circles: (1) a U-U mismatch on helix II and (2) the UGG motif on helix III. A single-stranded ring between helices I and IV is indicated with red left right arrows. It comprises the sequence GTACCC or *UACCC when T is substituted with U. The structures were directly folded and modelled using the ITS2-DB.

The most common S2 (Model 1) found in the *Ampelomyces* population is represented by a four-fingered model with a single-stranded ring rich in pyrimidines situated between helices I and IV. Model 1 also contains the conserved sequence, GTACCC (**Fig 5A**). In addition, between helices II and III there is another single-stranded ring of two nucleotides (A and T). Helix III comprises the largest helix with five internal loops that terminates in a single-stranded loop of six unpaired nucleotides.

Variations of Model 1, referred to as 1–2 and 1–3, were observed in two *Ampelomyces* sequences extracted from *A. mougeotii* (**Slides 1–3 in S2 File**). The variations included a deletion of a pyrimidine in an internal loop of helix II and pyrimidine additions in the single-stranded rings between helices II and III as well as III and IV (**slide 3 in S2 File**). Model 1 was found in those *Ampelomyces* spp. infecting several PM species representing seven genera, specifically from *A. mougeotii* on *Lycium hamilifolium, Erysiphe alni, Erysiphe berberidis, Erysiphe convolvuli, Erysiphe cruciferarum, Erysiphe euonymi, Erysiphe heraclei, Erysiphe polygoni, Erysiphe sordida, Erysiphe* sp., *Erysiphe trifolii, Golovinomyces cichoracearum* on *Aster salignus* and on *Lactuca* sp., *Golovinomyces orontii, Neoerysiphe galeopsidis, Oidium* sp., *Phyllactinia fraxini,* and *Podosphaera pannosa* (**slides 4–13 in S2 File**).

A major variation of Model 1 referred to as Model 7 was found from an ITS2 S2 from *Ampelomyces* sp. extracted from *A. mougeotii* infecting *L. hamilifolium* and sampled in Biatorbágy, Hungary (**Fig 5B**, **slide 14 in S2 File**). Model 7 contained transition and transversion mutations on helices III and IV, respectively. The common features of Models 1 and 7 are the presence of the U-U mismatch on helix II and the UGG motif on helix III.

The second most common S2, Model 2, (**Fig 6A**) contains a distinctive helix II that has two bulges ending in a large single-stranded loop of 12 unpaired nucleotides (**slide 14 in S2 File**). Between helices II and III there is a single ring with one nucleotide (A). In addition, helix III

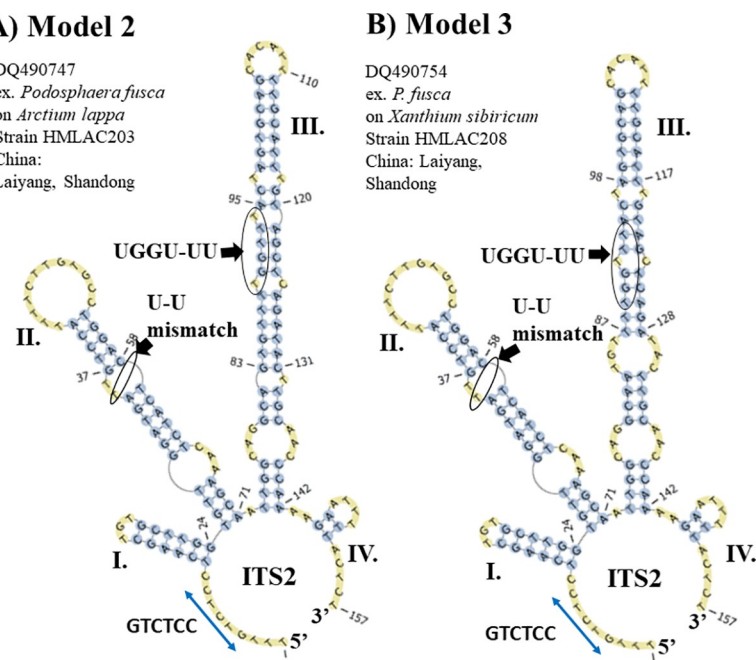

**Fig 6. The second most common ITS2 S2 (Model 2) found in the *Ampelomyces* population.** (A) The ITS2 S2 Model 2 derived from mycoparasites extracted from the three PM genera *Arthrocladiella*, *Golovinomyces* and *Podosphaera*. (B) The ITS2 S2 Model 3 was only found in one ITS2 sequence (GenBank accession number DQ490754) from *Ampelomyces* sp. extracted from *P. fusca*. The helices are indicated by Roman numerals (I–IV). Single-stranded rings (between helices I and IV) for Models 2 and 3 with the sequence GTGTCC are indicated with blue left right arrows. The following shared ITS2 features are indicated by black arrows and circles: (1) a U-U mismatch on helix II and (2) the UGGU and UU motifs on helix III. The structures were directly folded and modelled with the ITS2-DB.

has three internal loops, and two bulges with one unpaired nucleotide on each strand. A single-stranded ring with two nucleotides (A-A) is located between helices III and IV. In addition, Model 2 contains a single-stranded ring between helices I and IV that contains the highly conserved sequence, GTCTCC.

A Model 2 variant, referred to as Model 3, contains a major variation (**Fig 6B**) in helix III, which comprises four internal bulges. Model 3 is derived from an *Ampelomyces* ITS2 sequence (GI: DQ490754) extracted from *Podosphaera fusca* infecting *Xanthium sibiricum*, which was sampled in China. Model 2 was found in *Ampelomyces* ITS2 sequences isolated from the three PM genera, *Arthrocladiella*, *Golovinomyces* and *Podosphaera* (**slides 14–19 in S2 File**).

In addition, models 2 and 3 contain the U-U mismatch in helix II and the UGGU and UU motifs in helix III (**Fig 6A and 6B**).

On the other hand, the three ITS2 S2 Models 4–6 (**Fig 7A**–**7**C) were only found in *Ampelomyces* ITS2 sequences isolated from *P. leucotricha*, *Podosphaera ferruginea* and *E. necator* (section *Uncinula*), respectively (**slides 20–22 in S2 File**). Models 4–6 contain an internal loop in helix I and a single-stranded ring between helices II and III that is rich in purines (**Fig 7A**–**7**C). Between Models 4 and 5 (**Fig 7A and 7B**, **respectively**), major variations were observed in the internal loops and bulges of helix III. Model 6 was only found in *Ampelomyces* extracted from *E. necator*. The helix IV in Model 6 contains a stack of three paired nucleotides and terminates in a single-stranded loop of three unpaired Cs (**Fig 7C**). Unlike Models 1–3 and 7, Models 4–6 contain a single-stranded ring between helices II and III that is rich in As and a large helix III with a UGGU motif.

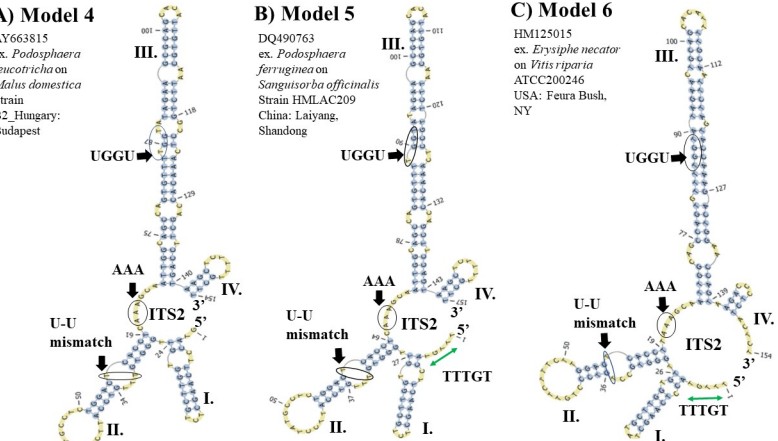

**Fig 7. Predicted ITS2 S2 found in the *Ampelomyces* population isolated from two PM genera, *Podosphaera* and *Erysiphe*.** (A) The ITS2 S2 Model 4 predicted from *Ampelomyces* spp. strains extracted from *P. leucotricha* on *Malus domestica*. (B) The ITS2 S2 Model 5 from *P. ferruginea* on *Sanguisorba officinalis*. (C) The ITS2 S2 Model 6 from *E. necator* infecting *Vitis* sp.. *M. domestica* and *S. officinalis* are plants from the family Rosaceae, whereas *Vitis* sp. is a plant from the family Vitacaee. The U-U mismatch motifs, a single-stranded ring rich in adenine nucleotides between helices II and III and a large helix III with a UGGU motif are indicated with black arrows and circles. The structures were directly folded and modelled with the ITS2-DB. The helices are indicated with Roman numerals (I–IV). The single-stranded ring between helices I and IV is indicated with green left right arrows. However, the one from Model 4 is not complete and the green arrow is not shown.

Conversely, most of the ITS2 sequences from *Ampelomyces* strains extracted from the PM *Erysiphe* genus had an ITS2 S2 represented by Model 1, except for those mycoparasitic strains extracted from *E. necator*, which comprise the distinctive ITS2 S2 Model 6 (**Fig 7C**). Minor changes in Model 6 were observed for *Ampelomyces* strains extracted from *E. necator*. These changes included transition mutations (C ↔ T) in the internal loop and hairpin of helix II and in the single-stranded ring between helices II and III (**slides 23–26 in S2 File**).

## Characterization of the ITS2 S2 improved environmental sequencing analysis

The predicted S2 of the putative *Ampelomyces* spp. from Groups 1 and 2 had the same four-fingered model, but the largest helix (helix III) contains between three and four internal loops and four or six bulges (**Fig 8A**–8**E**). In addition, all these S2s only contain the U-U mismatch motif on helix II. Unlike the ITS2 S2 from *Ampelomyces* spp. *sensu stricto*, S2 models from putative *Ampelomyces* contain a single-stranded ring between helices I and IV with the sequence, TCCATG. No variations in the hybridization model of the proximal stem were observed in fungal Group 1, **Table 4.1 in S1 File**. The ITS2 sequences from the putative *Ampelomyces* Group 1 isolated from plant material or from PM fungi were correctly annotated using both the ITS2-DB and homology modelling (**Table 4.2 in S1 File**).

The sequences from *A. humuli* (AF035779) and *A. quercinus* (AF035778) (**Fig 8A**) were homology modelled with the S2 of *D. glomerata* [GenInfo Identifier (GI) model: 332002412] and exhibited 99.4% similarity. Another sequence from an endophytic fungi referred to as *A. humuli* DQ093657 (**Fig 8B**) was homology modelled based on the GI model 226933708 of the ITS2 from *Didymella pomorum* formerly known as *Peyronellaea pomorum* [36]. This analysis revealed 98.8% identity between ITS2 sequences of *A. humuli* and *D. pomorum*. In addition, the homology modelling of *A. humuli* (KU204751), **Fig 8C**, based on the GI model 296840485 of the ITS2 from *Epicoccum* sp. ASR-245 revealed a 98.8% identity.

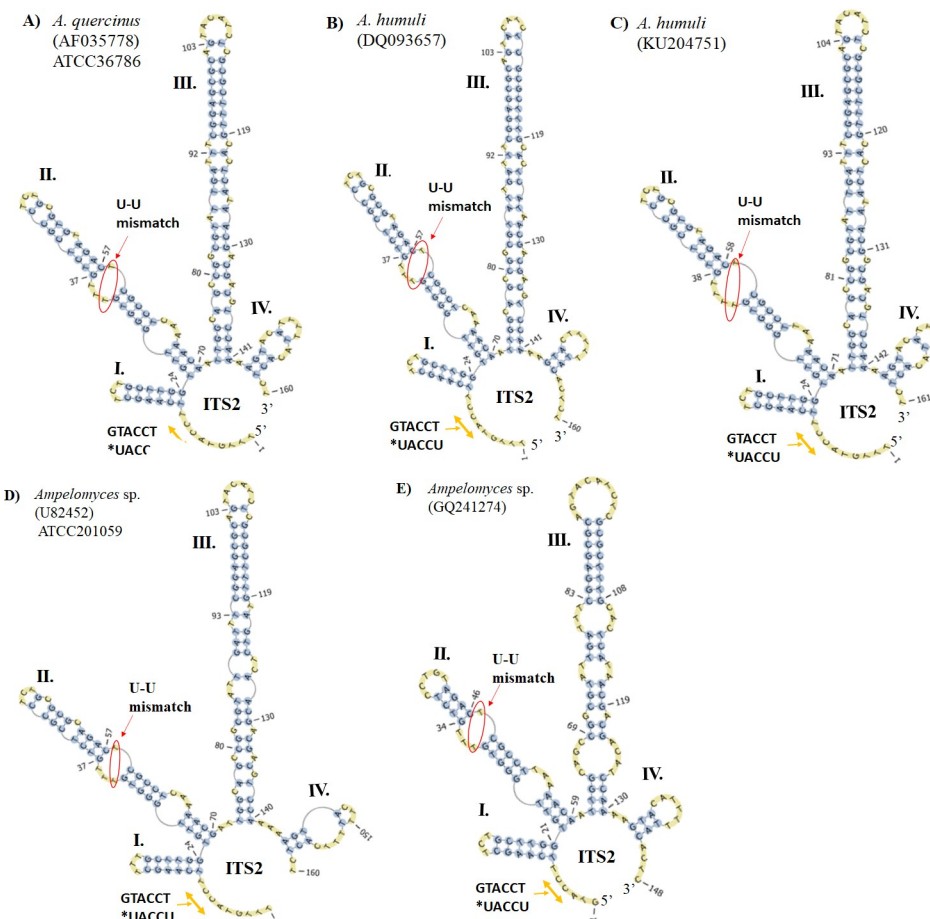

**Fig 8. Predicted putative *Ampelomyces* ITS2 S2s were modelled to those from *Didymella*, *Epicoccum* and *Phoma*.**
(A) The ITS2 S2 from *A. quercinus* was homology modelled based on the ITS2 S2 from *D. pomorum* (FJ839851). (B) The ITS2 S2 from *A. humuli* were modelled according to the ITS2 S2 from *Epiccocum* sp. (GU973791) and (C) from the ITS2 S2 from *D. glomerata* (FJ839851). (D) The ITS2 S2 from putative *Ampelomyces* sp. (Group 1) was homology modelled based on the structure of *P. labilis* (GU237868). (E) The ITS2 S2 from putative *Ampelomyces* sp. (Group 2) extracted from crosstie waste was directly folded and modelled. The helices are indicated with Roman numerals (I–IV). The single-stranded ring between helices I and IV is indicated by orange left right arrows. The U-U mismatch on helix II is indicated with red arrows and circles. The S2s were obtained with the ITS2-DB.

The last ITS2 sequences from Group 1 belonged to an *Ampelomyces* strain (U82452) extracted from *G. cichoracearum* infecting *Cucurbita pepo* (**Fig 8D**).This sequence was modelled with the structure of *Phoma labilis* (GI model: 294346519) and exhibited 98.8% identity. These results indicated that the Group 1 fungi are not related to the *Ampelomyces* lineages, but are related to the genera *Didymella*, *Epicoccum* and *Phoma*.

For Group 2 fungi (**Table 5.1 in S1 File**), only three ITS2 sequences were amenable for predicting the hybridization model of the proximal stem. These sequences are from putative *A. humuli* (AF455498, AF455518 and KT363070) and no variations in the hybridization model of their proximal 5.8S and 28S stem were detected. In addition, their S2s were not amenable to direct folding and instead a Basic Local Alignment Search Tool (BLAST) was conducted to search for their homologous sequences and S2s. This analysis demonstrated that fungi identified as *A. humuli* (AF455518 and AF455498) from human nasal mucus were amenable for modelling of their S2 based on the structure of the fungus *Cumuliphoma omnivirens*. In contrast, the S2 from *A. humuli* (KT363070) derived from soil could be predicted using a model of

the ITS2 S2 from the species *D. glomerata* with 99.4% of identity (**Table 5.2 in S1 File**). Two other ITS2 sequences (GQ241274 and LN80895) from putative *Ampelomyces* spp., which were extracted from environmental DNA samples (treated crosstie waste and air, respectively), were not amenable to verification of the hybridization of the proximal 5.8S and 28S stem due to the short length of the 28S strand. Nevertheless, the S2 from the fungus named *Ampelomyces* sp. (GQ241274) and isolated from creosote-treated crosstie waste could be folded directly (**Fig 8E**). It has a distinct S2; helix III has only four internal loops and a large hairpin formed by 10 unpaired nucleotides.

On the other hand, in order to predict the ITS2 S2 of the fungus identified as *A. humuli* from air samples, we conducted a BLAST search. This ITS2 S2 could only be partially modelled using the ITS2 S2 from species of the *Phoma* genus as a template with 32.5% of coverage. In addition, the ITS2 S2 of Group 2 were homology modelled using the ITS2 S2 models from the two fungal genera, *Cumuliphoma* and *Phoma*. These results support the hypothesis that these fungi are not related to the genus *Ampelomyces*. In summary, the ITS2 S2 from putative *Ampelomyces* sp. were homology modelled or directly folded and exhibited lower negative energy values between -25.8 and -35.9 than those from *Ampelomyces* spp. *sensu stricto*, which were obtained by direct folding and exhibited energy values between -47.3 and -36.8. This was another difference found between the ITS2 S2s of both fungal groups. The ITS2 S2 of fungi from the outgroup taxa were modelled using the structures from species of the *Didymella* and *Phoma* genera. This suggested that these sequences belonged to fungi from the genera *Didymella* and *Phoma* (**Table 6.1 and 6.2 in S1 File**).

## The ITS2 sequence-structure from PM-free environments did not cluster with those from *Ampelomyces* lineages

We aligned a sample of 21 directly predicted ITS2 S2s from *Ampelomyces* spp. *sensu stricto* together with those extracted from plant tissues and creosote-treated crosstie waste. The ITS2 sequence U82452, included in this analysis, was originally identified as belonging to *Ampelomyces* and was isolated from *G. cichoracearum*. Even though this mycoparasite co-infects the same PM host, this sequence was identified in early studies as belonging to fungi of the genus *Phoma*.

The evolutionary relationship of these fungi was estimated using the maximum likelihood method. The highest log likelihood of the ITS2 S2-based tree was -866.33 (**Fig 9A**) and the *Ampelomyces* population was grouped into three main clades. The first clade 1a contains all *Ampelomyces* sequences extracted from PM of the genus *Podosphaera* and it is represented by the ITS2 S2 Model 2 as well as its variations. Clade 1b contains *Ampelomyces* sequences extracted from *E. necator* on grapes and it is represented by the ITS2 S2 Model 6 and its variations. All clade 1b samples were collected in the USA. A very well-supported and independent clade 2 was formed by *Ampelomyces* sequences extracted from *P. leucotricha* from several parts of Europe and from *P. ferruginea* sampled in China. This clade 2 is represented by the ITS2 S2 Models 4 and 5. Clade 3 consists of *Ampelomyces* sequences that were isolated from *A. mougeotii* in different locales in Hungary and it is represented by the ITS2 S2 Models 1 and its variations as well as Model 7. Conversely, similar to the ITS2 S2 sequence from *P. herbarum* (GI JF810528), fungi strains isolated from *G. cichoracearum*, *P. abies* decayed root and creosote-treated crosstie waste were not grouped into clades 1–3. This result is in agreement with our previous findings, which confirm that these environmental DNA sequences are not related to the *Ampelomyces* genus.

We also compared the previous tree (**Fig 9A**) with the one based on the complete ITS region (**Fig 9B**). We found similar results. However, clade 2a emerged as a sister clade of clade

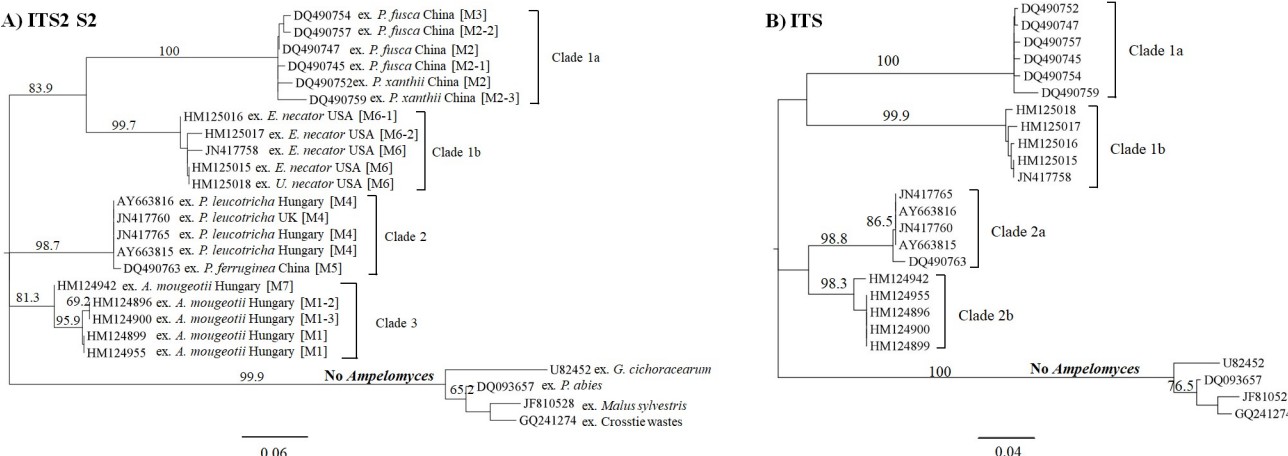

**Fig 9. Phylogram based on ITS2 S2s enhance the discrimination between fungal DNA environmental samples and the 'true' *Ampelomyces*.** (A) The phylogram with the highest log likelihood (-866.33) was based on the ITS2 S2s and inferred via the maximum likelihood (ML) method and the Kimura two-parameter model. An evolutionary rate among sites was modelled with a discrete gamma distribution (+G) parameter = 0.99. (B) The phylogenetic tree with the highest log likelihood (-2040.24) was based on the ITS region and estimated with the maximum likelihood method and the Tamura-Nei model with a discrete gamma distribution (+G) parameter = 0.48. The ML bootstrap values >60% are indicated over the branches and are expressed as percentages. The scale bar represents the nucleotide substitutions per site. The tree was edited with FigTree v1.4.4 software. The GenBank accession numbers are indicated before the taxa names. Abbreviations: ITS2 S2 variations of main Models 1 (M1, M1-2 and M1-3); Models 2 (M2, M2-1, M2-2 and M2-3); Model 3 (M3); Model 4 (M4); Model 5 (M5); Models 6 (M6-1 and M6-2); and Model 7 (M7).

2b. Among the outgroup taxa samples, the ITS region resolved better than the ITS2 S2. Part of this could be due to the small number of samples used in the ITS2 S2 analysis. Even though clades 1a and 3 were represented by *Ampelomyces* extracted from PMs of the genera *Podosphaera* and *Arthrocladiella*, respectively, it does not indicate that these S2s are mycohost-associated. Indeed, in the clade of *Podosphaera*, other Model 2 sequences from *Ampelomyces* extracted from other PMs such as *Arthrocladiella* and *Golovinomyces* were grouped together (**Fig 10**). Nevertheless, no additional S2 Models (2 nor 3) were found in those samples from Europe. Instead, they were only found in China. Conversely, the ITS2 S2 Model 1 was grouped with others from *Ampelomyces* extracted from seven PM genera.

In order to support our hypothesis that the ITS2 S2 analysis can improve the differentiation between the 'true' *Ampelomyces* and unrelated fungi, we determined the genetic divergence distances among groups between the phylogenies. These distance calculations were based on the simultaneous alignment of the ITS2 S2 (**Table 4**) and the complete ITS region (**Table 5**). For the ITS2 S2 analysis, the maximum distance observed between clades 1a and 3 was 0.298 ± 0.056 standard error (S.E.) while those between the outgroup taxa and clade 3 was 0.403 ± 0.073 (S.E.), (**Table 5**).

In contrast, by using the ITS-like DNA barcode, we found that the maximum distance between closed related *Ampelomyces* clades 1a and 2b was 0.186 ± 0.021 (S.E.), while the minimum distance between the outgroup and an *Ampelomyces* clade (clade 2a) was 0.195 ± 0.021 (S.E.), (**Table 5**). These findings suggested that the ITS could not distinguish *Ampelomyces* from other closely related fungal genera, while the ITS2 S2 exhibited greater discrimination between the outgroup taxa and the closely related *Ampelomyces* clade 3.

Other differences were observed when comparing all consensus ITS2 S2s belonging to each clade (**Fig 11A–11D**). All *Ampelomyces* consensus structures contain the U-U unpaired motif in helix II. In helix III, the UGG and UU motifs in Models 2 and 3 (**Fig 11A**), the UGGU motif in Models 6 (**Fig 11B**), 4 and 5 (**Fig 11C**) and the UGG motif in Models 1 and 7 (**Fig 11D**) formed the clades 1a, 1b, 2 and 3, respectively. The AAA motif between helices II and III was

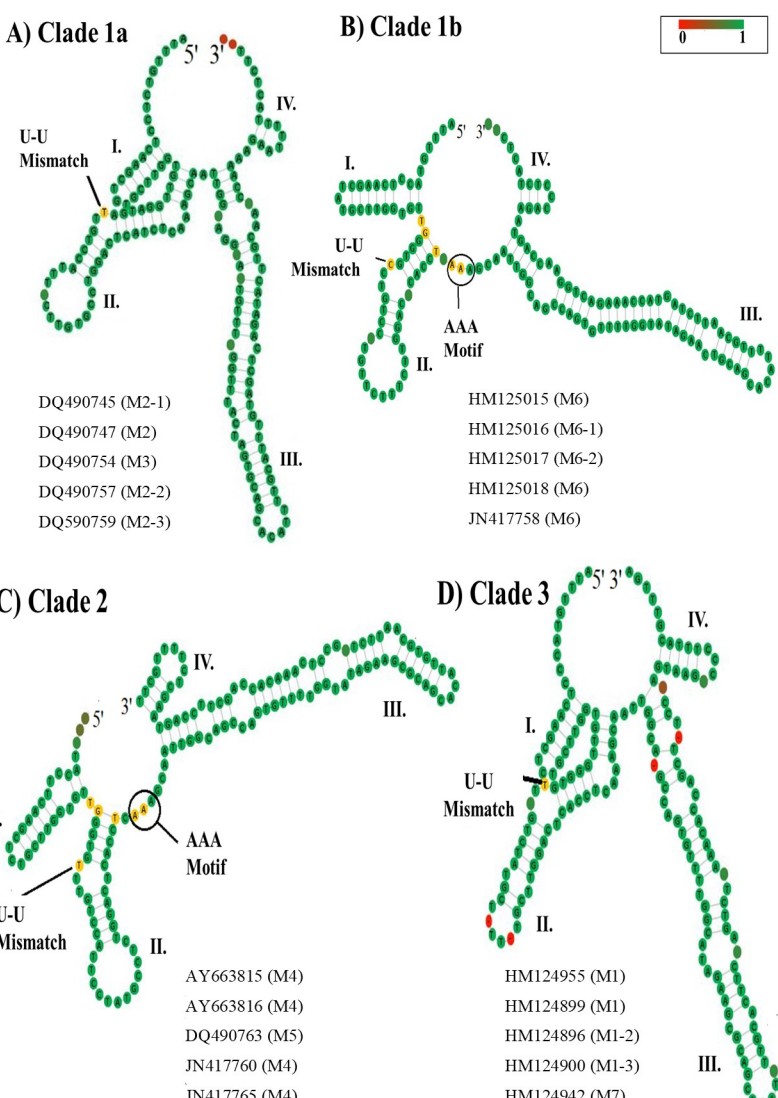

**Fig 10. Phylogram based on 26 *Ampelomyces* ITS2 S2s shows that the S2s are not associated with PM hosts.** The phylogram with the highest log likelihood (-829.62) is shown based on the Kimura two-parameter DNA model and an evolutionary rate among sites modelled with a discrete gamma distribution (+G) parameter = 0.90. The ML bootstrap values >60% are indicated over the branches and are expressed as percentages. The scale bar represents the nucleotide substitutions per site. The tree was edited with FigTree v1.4.4 software. The GenBank accession numbers are indicated before the taxa names. Abbreviations: ITS2 S2 variations of main Models 1 (M1, and M1-3); Models 2 (M2, M2-1 and M2-3); Model 3 (M3); Model 4 (M4); Model 5 (M5); Models 6 (M6-1 and M6-2); and Model 7 (M7).

only present in *Ampelomyces* extracted from *E. necator* in clade 1b (**Fig 11B**) and in *Ampelomyces* extracted from *P. leucotricha* and *P. ferruginea* in clade 2 (**Fig 11C**).

Conversely, the consensus structure from the outgroup taxa do not contain the AAA motif between helices II and III but it does harbor the unpaired U-U motif in helix II (**Fig 12A**). When the predicted consensus ITS2 S2 from the outgroup was built based on the alignment of the four ITS2 sequences and structures including the gaps (**Fig 12B**), the consensus S2 was highly modified, indicating that these sequences belong to diverse fungi.

After this comprehensive analysis, the selected nucleotide sequences belonging to *Ampelomyces* spp. *sensu stricto* were included in the **S3 File**.

**Table 4. Estimates of evolutionary divergence over sequence pairs between fungal groups when analysing the ITS2 sequences and structures.**

| ITS2 | 1 | 2 | 3 | 4 | 5 |
|---|---|---|---|---|---|
| **1. Clade 1a** | | 0.045 | 0.050 | 0.056 | 0.090 |
| **2. Clade 1b** | 0.238 | | 0.049 | 0.042 | 0.079 |
| **3. Clade 2** | 0.275 | 0.235 | | 0.033 | 0.080 |
| **4. Clade 3** | 0.298 | 0.204 | 0.137 | | 0.073 |
| **5. Outgroup taxa** | 0.533 | 0.477 | 0.444 | 0.403 | |

The simultaneous ITS2 S2 alignment was used to calculate the genetic distances among groups via the log-det (Tamura-Kumar) method using MEGA-X V10.1.8 software. The values represent the number of base substitutions per site obtained by averaging over all sequence pairs between groups. The standard error estimates are shown in blue.

## Discussion

Improving the accuracy in identifying the ITS sequences derived from environmental DNA sampling of fungi is expected to benefit future fungal genetic population studies including those for *Ampelomyces*. In addition, it may also provide insight into the mechanisms of parasitism, which may improve the use of biocontrol agents against PMs. Previous researches have shown that fungi identified as *A. humuli* or *A. quercinus* were not related to the mycoparasites *Ampelomyces* based on their culture characteristics and phylogeny [30,31]. Indeed, this research shows ITS sequences from plant tissues, environmental DNA samples and human mucus do not belong to species of the genus *Ampelomyces*. This classification is based on their ITS sequence length and nucleotide content, their ITS2 S2 together with simultaneous ITS2 sequence-structure alignment and its corresponding maximum likelihood tree.

On the other hand, S2 analysis has provided insight into the regulatory mechanisms of mycoparasite pre-RNA S2. Based on the phylogenetic tree of ITS2 sequences and structures, we propose that a basic ribosomal regulatory mechanism exists across the entire mycoparasite population since the S2 Model 1 was observed in strains extracted from seven PM genera. Moreover, major differences in S2 were found in the internal loops of hairpins that are essential for alternative splicing during pre-RNA processing [74]. Indeed, mutations occurring adjacent to these areas are known to stop the rRNA maturation process [75]. Interestingly, only ITS2 S2 Models 4–6 from *Ampelomyces* strains isolated from *P. leucotricha*, *P. ferruginea* and *E. necator*, respectively, share conserved motifs i.e., poly adenine nucleotides in the single-stranded ring between helices II and III, that were previously reported in yeasts and vertebrates [73]. This suggests that in some *Ampelomyces* mycoparasites, post-transcriptional processes for ribosomal biogenesis may be similar across higher organisms, such as fish and mammals [73]. Conversely, ITS2 S2 models 1–3 and 7 that do not have this conserved motif may have different regulatory mechanisms.

**Table 5. Estimates of evolutionary divergence over sequence pairs between fungal groups when analysing the ITS region.**

| ITS | 1 | 2 | 3 | 4 | 5 |
|---|---|---|---|---|---|
| **1. Clade 1a** | | 0.0197 | 0.0189 | 0.0206 | 0.0265 |
| **2. Clade 1b** | 0.1659 | | 0.0125 | 0.0203 | 0.0247 |
| **3. Clade 2a** | 0.1561 | 0.0798 | | 0.0183 | 0.0213 |
| **4. Clade 2b** | 0.1861 | 0.1717 | 0.1532 | | 0.0263 |
| **5. Outgroup taxa** | 0.2733 | 0.2287 | 0.1948 | 0.2548 | |

The values represent the number of base substitutions per site obtained by averaging over all sequence pairs between groups. The distances were determined via the log-det (Tamura-Kumar) method using MEGA-X V10.1.8 software. The standard error estimates are shown in blue.

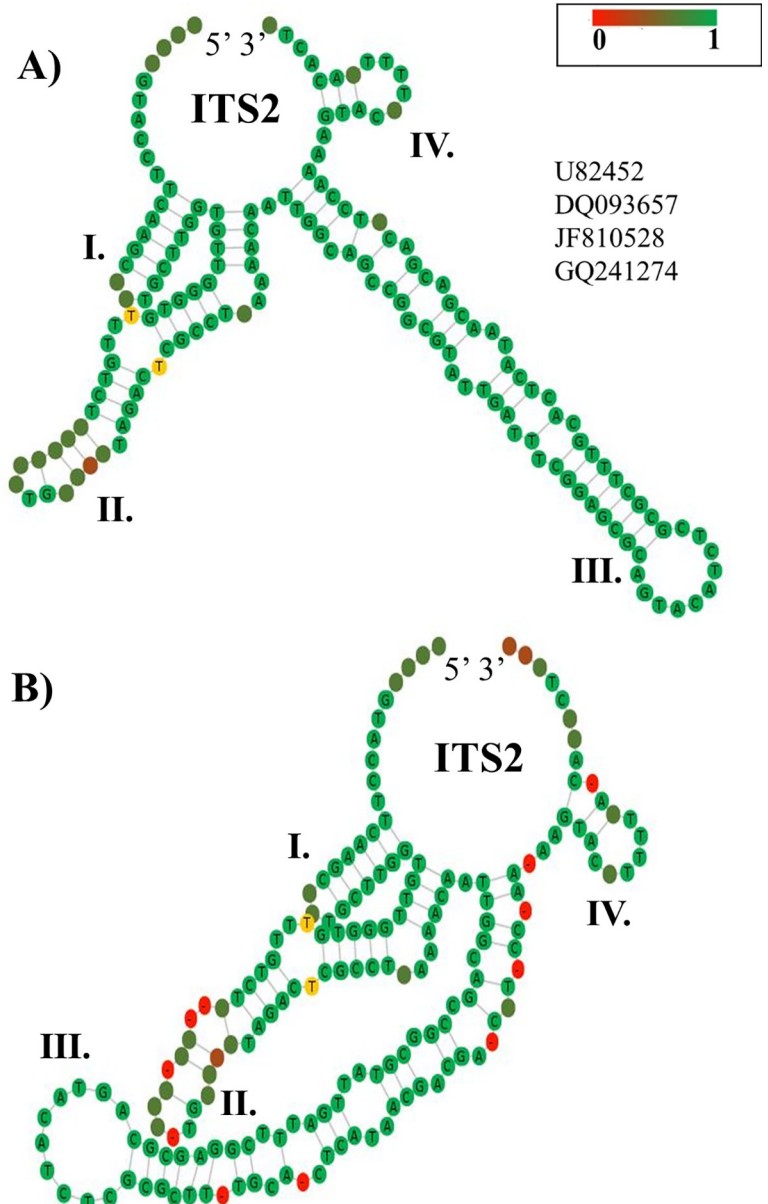

**Fig 11. Four major consensus ITS2 S2s from *Ampelomyces* are represented in three main clades.** (A) Consensus ITS2 S2 of Models 2 and 3, which are grouped in clade 1a. (B) Consensus ITS2 S2 belonging to clade 1b are represented by Model 6. (C) Consensus ITS2 S2 of Models 4 and 5 that belong to clade 2. (D) Consensus ITS2 S2 of Models 1 and 7 that belong to clade 3. All consensus ITS2 S2s have gaps and were obtained with the 4SALE v1.7.1 software. Yellow nucleotides indicate conserved motifs (U-U mismatch and AAA motifs) known to be found in yeasts and vertebrates [73]. UGG, UGGU and GGUU motifs on helix III are indicated with black circles. The bar on the upper right side indicates the level of nucleotide conservation, with the most conserved nucleotides in green. The gaps are depicted in red. The GenBank IDs used to build each consensus are indicated below each S2.

Early studies demonstrated that when using ITS sequences and microsatellites as DNA barcodes, the only haplotype found in *Ampelomyces* extracted from *P. leucotricha* on apples (APM *Ampelomyces*), which caused epidemics in the spring, may be the result of its PM host phenology [23]. However, there was an exception to this finding; APM *Ampelomyces* grouped with non-APM *Ampelomyces* (isolated from *P. ferruginea* and *P. pannosa*). Moreover, isolation

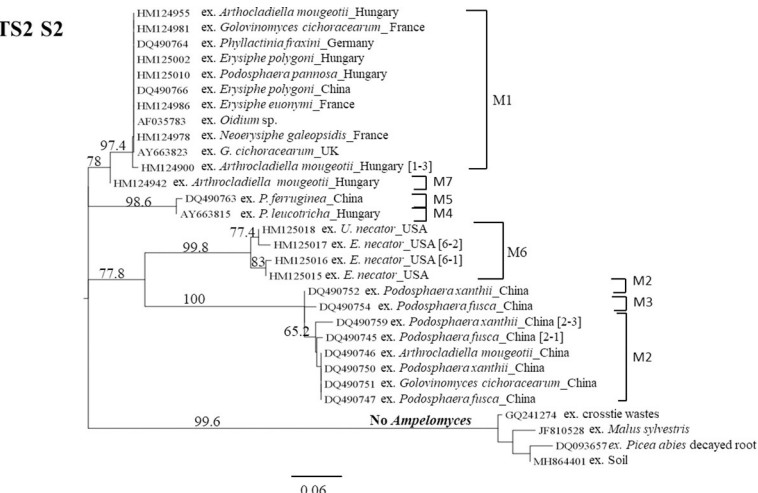

**Fig 12. Highly diverse consensus ITS2 S2s from the outgroup taxa.** (A) The consensus ITS2 S2 was elaborated without gaps. The GenBank ID numbers of four ITS2 sequences used to predict the ITS2 S2 are indicated. (B) The consensus ITS2 S2 build with gaps. The consensus ITS2 S2s were obtained using the 4SALE v1.7.1 software. The helices are indicated by Roman numerals (I–IV). Nucleotides in yellow indicate the U-U mismatch motifs in helix II. The bar on the upper right side indicates the nucleotide conservation across the four ITS2 sequences used to obtain the consensus structure. Gaps in the structure are indicated in red.

date of one of the strains was unknown, thus the impact of mycohost phenology on the evolution of the only *Ampelomyces* ITS haplotype remains unclear. In our phylogenetic study based on ITS2 sequence and structure, we obtained clustering patterns similar to those reported previously, but in our case each clade indicated that *Ampelomyces* can be grouped into four major S2s. This also explains why non-APM *Ampelomyces* clustered with APM *Ampelomyces*. These different structures also imply that different mechanisms of ribosomal regulation may occur.

These results beg the question: how did these different ITS2 S2s originate? In a previous study, it was proposed that the four major ITS2 S2s found in the ITS sequences of the hyperparasitic fungus *Sphaeropsis visci* may be due to sexual recombination, although no teleomorphs were found [35]. Microsatellite analysis of APM *Ampelomyces* [23] showed that they undergo frequent asexual reproduction and regular recombination, and an *Ampelomyces* sexual reproductive state was not observed. Interestingly, unequal crossing over occurs during mitosis in other fungi such as *Saccharomyces cerevisiae* [76] and *Ceratocystis manginecans* [77]; we cannot discount its occurrence in *Ampelomyces*, whose sexual reproductive stages were not detected in the early studies. It is also possible that the ITS2 S2 found in APM *Ampelomyces* in Europe is the result of a recombination process with other non-APM *Ampelomyces*. However, this S2 was also observed in *Ampelomyces* extracted from other PMs, such as *P. ferruginea* and was found in other locales, such as China. Another explanation is derived from a previous report [78], which demonstrated that specific regions of the rRNA cistron in humans and several primates maintain polymorphic sites by natural selection. Since the ITS2 C/G content and minimum free energy values of ITS2 structure formation were not significantly different to those from the rest of the population (data not shown), we hypothesize that the variations observed in the proximal 5.8S and 28S stem indicate an early stage of pseudogene formation in two *Ampelomyces* ITS sequences extracted from *U. necator* and *P. xanthii*. The polymorphisms observed in the proximal 5.8S and 28S stem and the ITS2 can be either homogenized across the cistrons by unequal crossing over [76] or gene conversion through concerted evolution, and their fate determined by genetic drift. Nevertheless, the modifications observed in the proximal 5.8S and 28S stem may affect positively or negatively ITS2

processing. For example, the first variation of the hybridization model, an ITS2 derived from an *Ampelomyces* strain extracted from *P. xanthii* contains a short proximal 5.8S and 28S stem. This short stem may affect the enzymatic machinery that processes the ITS2 because the length of the stem is important for processing [75]. Consequently, the mutations observed in both strands 5.8S and 28S may be eliminated by genetic drift through concerted evolution [77,79]. On the other hand, the second variation of the hybridization model, an ITS2 derived from an *Ampelomyces* isolated from *U. necator*, may have a more relaxed structure that is still functional. The function of the secondary molecules resulting from the processing of the ITS2 remains to be elucidated. In addition, it remains to be determined whether these secondary molecules provide a specific trait to the fungus. Indeed, it has recently been shown that ribosomes containing rRNA variants have altered gene expression and physiology [80]. If the ITS sequence becomes a ghost pseudogene that favours the synthesis of proteins via ribosomal regulation, then it may be naturally selected by inhabiting a new environment under different biotic or abiotic pressures. Based on this, we think that the random appearance of a ghost pseudogene may lead to the formation of new individuals that form sister clades. For instance, in the phylogram of ITS2 S2s, clade 2 will be divided into two sister clades where each is formed by APM *Ampelomyces* and non-APM *Ampelomyces* ITS2 S2s. This evolutionary process may be initiated by the random appearance of a ghost pseudogene in ITS sequences of APM *Ampelomyces* that was selected by phenology of its powdery mildew host. Nevertheless, no additional samples were available from *Ampelomyces* isolated from *P. ferruginea* in China; and this needs to be addressed in the future.

On the other hand, the interspecific genetic distances between the 'true' *Ampelomyces* -containing clades and the whole outgroup taxa were higher than the intraspecific genetic divergences among the 'true' *Ampelomyces* clades calculated by using either the ITS2 S2s alignment or the ITS alignment. However, the barcode gap was notably enhanced by using the ITS2 S2s. This demonstrates that ITS2 S2s are an important tool to enhance the identification of *Ampelomyces* mycoparasites from DNA environmental samples.

Further research is required in this area together with sampling across different crops, countries and seasons. Partial sequences of the ITS region deposited in GenBank and belonging to *Ampelomyces* spp. are valuable, but we can obtain more information when whole sequences are used. We propose that future studies publish complete *Ampelomyces* ITS sequences and sample the same host at least two times.

In summary, we found that ITS sequences from fungi derived from PM-free environments are still being deposited in the GenBank database under the generic name of *Ampelomyces*. We also showed that these sequences are not related to the genus *Ampelomyces* based on their ITS sequence length, nucleotide composition and the simultaneous alignment of their ITS2 sequences and structures. Finally, we detected for the first time that pseudogene formation could occur in the nrDNA ITS region of *Ampelomyces* mycoparasites. Our study suggested that complete ITS2 sequences are crucial for understanding the phylogeny of these understudied *Ampelomyces* mycoparasites.

## Conclusions

In this study, we demonstrated the utility of ITS2 S2 analysis to unveil underlying evolutionary processes in the *Ampelomyces* mycoparasites. Furthermore, the predicted S2s represent a valuable source of information that enhanced the analysis of environmental sequencing and advanced our knowledge in the field of fungal genetic population biodiversity. We highly recommend using the ITS2-DB for this purpose. On the other hand, the controversy surrounding whether the ITS region alone is suitable to discriminate between closely related species may be

ameliorated by analysing S2 together with phylogenetic studies based on simultaneous sequence-structure alignment of the ITS2s. Our results support the utility of this strategy as well as the view that the ITS region is an excellent primary fungal barcode marker, and we are beginning to uncover its real potential.

## Supporting information

**S1 File. DNA ITS and ITS2 secondary structures sequence analyses.**
(ZIP)

**S2 File. ITS2 secondary structures from *Ampelomyces* spp. *sensu stricto* and putative *Ampelomyces* spp.**
(ZIP)

**S3 File. Datasets.**
(TXT)

## Acknowledgments

We thank Dr. Douglas Eacersall from USQ and the reviewers for their valuable suggestions.

## Author Contributions

**Conceptualization:** Rosa E. Prahl.

**Data curation:** Rosa E. Prahl.

**Formal analysis:** Rosa E. Prahl, Shahjahan Khan, Ravinesh C. Deo.

**Supervision:** Shahjahan Khan, Ravinesh C. Deo.

**Writing – original draft:** Rosa E. Prahl.

**Writing – review & editing:** Shahjahan Khan, Ravinesh C. Deo.

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
