## [Decision Letter · Decision Letter 0]

7 Jan 2021

PONE-D-20-39142

ITS2 Secondary Structures in Mycoparasitic Ampelomyces: Evolution and Environmental DNA Analysis

PLOS ONE

Dear Dr. Prahl,

Thank you for submitting your manuscript to PLOS ONE. After careful consideration, we feel that it has merit but does not fully meet PLOS ONE’s publication criteria as it currently stands. Therefore, we invite you to submit a revised version of the manuscript that addresses the points raised during the review process.

The paper is interesting and generally the work performed well. Howevr, according to the reviewer's suggestions, it needs major modifications in order to render it acceptable for publication.

We look forward to receiving your revised manuscript.

Kind regards,

Sabrina Sarrocco

Academic Editor

PLOS ONE

Journal Requirements:

Reviewers' comments:

Reviewer's Responses to Questions

**Comments to the Author**

1. Is the manuscript technically sound, and do the data support the conclusions?

Reviewer #1: Partly

2. Has the statistical analysis been performed appropriately and rigorously? 

Reviewer #1: Yes

3. Have the authors made all data underlying the findings in their manuscript fully available?

Reviewer #1: Yes

4. Is the manuscript presented in an intelligible fashion and written in standard English?

Reviewer #1: No

5. Review Comments to the Author

Reviewer #1: The manuscript by Rosa Prahl and co-authors describe a computational study of the importance of the secondary structure of ITS2 for identification of Ampelomyces strains in environmental samples. Overall, the work is relevant and performed with appropriate methods. However, it lacks in clarity of the presentation and in the interpretation and discussion of parts of the results.

1. The abstract is not providing enough information to understand the significance of the work. Why is molecular identification of fungi based on environmental DNA challenging? Due to low levels of polymorphisms in different markers, or too much variation etc., please be specific. Why did you predict ITS2 secondary structures? No explanation of the significance is provided. It is too bold to state that the polymorphisms identified are key drivers of ITS evolution, the mutations are the result from an evolutionary process, hardly the drivers.

2. It is difficult to follow the significance and purpose of the different steps in the Result section. This may partly be due to the format where materials and methods comes after the discussion, but it may help to provide a little more info regarding this in the beginning of the different sections in the results.

3. The discussion concerning pseudogenes (lines 459-472) is vague and is some aspects problematic. The provided reference for pseudogenes (Harpke and Peterson) merely discuss this in the context of the 5.8S gene, not the ITS. This is understandable, as the term pseudogene is usually related with expression, and Harpke and Peterson indeed takes this into account by comparing 5.8S sequences from genomic DNA verses cDNA. It is however clear that the secondary structure of ITS is important for the splicing but whether the observed polymorphisms are enough to discuss loss-of-function paralogs is not clear to me. This part of the discussion needs to be made more specific. The last part of this section (line 466 onwards) is difficult to understand. Why would these pseudogenes be naturally selected? It would be more reasonable to assume release of selective constraints, leading to continuous accumulation of mutations, with the ultimate fate determined by genetic drift. Also, the evolution of the rRNA repeat is very much depending on concerted evolution, but this is not even mentioned nor discussed. In fact, as concerted evolution at least partially can be connected with meiosis and sexual reproduction, it would make sense to discuss this in the current manuscript.

4. The comparison of the secondary structure prediction and the phylogenetic analysis is lacking from the discussion. The authors repeatedly state that the secondary structure prediction of ITS is a valuable tool for identifying Ampelomyces sequences from the environment. This may indeed be the case, but it seems to me that the phylogenetic analysis provides exactly the same result and resolution. What exactly is the gain from using secondary structure prediction? I am lacking a critical discussion of this in the manuscript.

5. The analysis of length variation. First, it would be useful to provide a statement regarding the distribution of the ITS1 length variation between Ampelomyces and the other groups, is it due to specific indels or more evenly spread out over the sequence? Secondly, what was the basis for defining groups 1 and 2? This was not clear to me. Third, why is the normalization of the length even necessary? The authors argue that it was necessary due to the use of different primers, but if all used sequences were complete full-length (ITS1-5.8S-ITS2) as stated, the primers should not matter. I assume that partial 18S and 28S sequences should be deleted before the analysis. This is not clear to me.

Minor issues:

1. Lines 41-43. The 5.8S gene is not assumed to evolve neutrally. Probably an error in the sentence.

2. Line 53. I would exchange “although only” with “and” in order to be a bit more positive concerning the application aspects. Commercialization is a complicated process, so it is great that two strains have gone the whole way to products.

3. Line 55. Delete “conidia, fruiting bodies”.

4. Line 97. Table 3 comes before Table 2.

5. Line 109. Replace “extracted” with “retrieved”.

6. Line 110. Replace “the mycoparasites” with “Ampelomyces mycoparasites”.

7. The figures were of poor quality, but it may be related with the review versions.

8. Line 274. “Major” should be “major”.

9. Line 282. “helices IIII” should be “helices III”, I guess.

10. Line 369. What is the significance of mentioning the free energy value for this structure but not for the others?

11. Line 410. Replace “from” with “with”.

12. Figure 9 + legend. Bootstrap support values are typically reported as the percentage, e.g. 53 instead of 0.53.

13. Line 446. “neighbor-joining” should be “maximum likelihood”, right?

14. Lines 450-452. Is the Bartys et al reference really appropriate here? The reference seems to deal with hairpin structures in mRNA, while the importance in non-coding RNA may be different.

6. PLOS authors have the option to publish the peer review history of their article (what does this mean?). If published, this will include your full peer review and any attached files.

Reviewer #1: No

---

## [Author Response · Author response to Decision Letter 0]

2 Mar 2021

Detailed Response to Reviewer 1

The authors would like to thank the Editor in Chief and Reviewer 1 for taking their time to read our manuscript, provide constructive criticisms and helpful comments. We have revised the manuscript accordingly. The point by point response to the comments of the Reviewer 1 are provided below. The subsequent changes in the manuscript are also highlighted to demonstrate the changes that were made in revised version.

Reviewer #1: The manuscript by Rosa Prahl and co-authors describe a computational study of the importance of the secondary structure of ITS2 for identification of Ampelomyces strains in environmental samples. Overall, the work is relevant and performed with appropriate methods. However, it lacks in clarity of the presentation and in the interpretation and discussion of parts of the results.

1. The abstract is not providing enough information to understand the significance of the work (1.1-1.4). 

A new abstract has been written (Lines 27-49).

1.1 Why is molecular identification of fungi based on environmental DNA challenging? 

Answer: Accurate fungal identification depends on the method used to classify the fungus. Several taxonomical and phylogenetic fungal groups require appropriate cultural conditions before identification and specific primers for successful amplification of molecular markers. This has been clarified in the revised version. Please see Lines 67-71.

1.2 Due to low levels of polymorphisms in different markers, or too much variation etc., please be specific. 

Answer: In accordance with the taxonomy and phylogeny of the fungal group under study, the use of only one molecular marker can result in poor species discrimination resolution (Schoch et al., 2012). For a barcode gap, the interspecific genetic variability needs to be higher than the intraspecific variability. In fungi, the nuclear ribosomal DNA (nrDNA) internal transcribed spacer (ITS) region has been widely used as a DNA barcode (Schoch et al., 2012). For fungal identification, the ITS region has enough genetic variability to the species level, but sometimes it does not resolve closely phylogenetic related species (Bruns, 2001) and some fungi require extra identifiers for species delimitation within a determined genus or family. The nrDNA ITS region is the preferred DNA barcode for fungi (Schoch et al., 2012). However, it can have some limitations. If some taxa have low ITS interspecific variability other molecular markers need to be used to precisely report genetic diversity (Gazis et al., 2011). Conversely, intragenomic variation caused by the presence of several nonorthologous and paralogous copies in a single conidium (Lindner and Banik, 2001; and Kovács et al., 2011) can result in overestimate the intraspecific variability in some fungi (Gomes et al., 2002 and Simon et al., 2008). For instance, in Glomeromycota, the upper range of this variability in the ITS region can reach up to 20% within a single spore (Krüger et al., 2012). This has been added to the revised manuscript. Please see Lines 71-86).

1.3 Why did you predict ITS2 secondary structures? No explanation of the significance is provided. 

Answer: This is a useful point to further increase the clarity of the revised manuscript. The following rationale has prompted us to predict Ampelomyces secondary structures (S2): (1) Interspecific divergences among some Ampelomyces groups have reach up to 15%; and extra markers may be used to accurate identify these species (2) most of the phylogenetic studies conducted on Ampelomyces were based on the nrDNA ITS region and the successful design of primers for other markers are difficult to obtain, which often occurs for other fungi and (3) the utility of ITS2 secondary structures (S2) of Ampelomyces have not been evaluated.

In order to resolve these issues and contribute to knowledge and future research in this area, we have investigated whether ITS2 secondary structures (S2) contains specific features that may be incorporated into phylogenetic analysis and provide with novel important information regarding evolutive processes occurring in Ampelomyces. Indeed, the utility of the ITS2 S2 to study the phylogeny of several organisms have been demonstrated (Gottschling et al. 2004 and Poczai et al. 2015) but it has not been investigated in Ampelomyces.

This has been added to the revised manuscript. Please see lines 110-122.

1.4 It is too bold to state that the polymorphisms identified are key drivers of ITS evolution, the mutations are the result from an evolutionary process, hardly the drivers.

Answer: Thank you for this comment. We agree with your comment and we wish to clarify some points. We focus on the polymorphisms found in the hybridization model of the proximal 5.8S and 28S stem and not on other polymorphic sites that occurred mainly in the bulges and single-stranded rings of the ITS2 S2.

According to the classification introduced by Zheng and Gerstein, some pseudogenes may have some functionality in RNA coding regions; and classified them as ‘ghost pseudogenes’. These ghost pseudogenes may regulate responses to environmental stresses in yeasts, while ‘dead pseudogenes’ do not have any function and may be subjected to evolutionary genetic drift. Variations in the hybridization model of the proximal 5.8S and 28S stem suggested that the formation of pseudogenes occurs in the nrDNA ITS region of Ampelomyces strains. Thus, our study, adding new knowledge to the body of research, has indicated for the first time that the formation of pseudogenes can occur in the nrDNA ITS region from two distinct Ampelomyces infecting cucumber and grape powdery mildews (PMs). Since the correct structure and assembly of the proximal 5.8S and 28S stem determines whether the enzymatic ribosomal machinery will continue to form mature 5.8S and 28S rRNAs (Côté et al. 2001), we hypothesize, that the modification observed in the proximal 5.8S and 28S stem from two Ampelomyces spp. may decrease the efficiency of ribosomal maturation or alternatively, it may lead to the production of other molecules that favour the maturation of rRNA for protein synthesis. In detail, the variations observed in the proximal 5.8S and 28S stem from Ampelomyces sampled from Podosphaera xanthii, indicated that the short length of the stem may impede the maturation of the pre-rRNA as previously shown (Côté et al. 2001) and consequently the mutations observed in both strands 5.8S and 28S may be eliminated by genetic drift through concerted evolution (Elder and Turner, 1995 and Naidoo et al., 2013). However, the second modification detected in the sequence from Ampelomyces extracted from Erysiphe necator (Section Uncinula) may provide some flexibility in the stem as reported by Côté et al. 2001. and could lead to the production of secondary molecules that positively affect ribosomal biogenesis. If the latter is the case, this could lead to the formation of a ghost pseudogene with a specific function. Thus, the ghost pseudogene may undergo natural selection. Therefore, in the absence of sexual reproduction, the random appearance of pseudogenes may lead to the formation of new individuals that form sister clades. 

On the other hand, compared to functional genes, pseudogenes have low C/G content and high sequence length variation. However, these characteristics in the two putative pseudogenes found in Ampelomyces were not different to those from the rest of the population. Interestingly, the minimum free energy value of the S2 formation of the 5.8S gene in one of the putative pseudogenes from Ampelomyces extracted from P. xanthii, was slightly less negative than those from 5.8S sequences belonging to Ampelomyces infecting the same or different PM hosts. This result indicated that this polymorphism may be a precursor in the early stages of pseudogene formation.

In addition, Keller et al. reported that variations in the hybridization model can naturally occur. Nevertheless, we do not believe this is the case in our study because amongst the observed variations in the proximal 5.8S and 28S stem S2, there were no predominant variants in the entire Ampelomyces population. 

We found reliable evidence that the formation of pseudogenes may occur randomly in the ITS loci from two Ampelomyces populations. What we meant by the word drivers is that ghost pseudogenes may have occurred first in any loci and if they improved the maturation of the pre-rRNA, then these pseudogenes will undergo natural selection caused by either temporal isolation or geographical isolation instead of evolving by neutral drift. For clarity, we modified this section in the Discussion section. Please see Lines 827-850.

2. It is difficult to follow the significance and purpose of the different steps in the Result section. This may partly be due to the format where materials and methods comes after the discussion, but it may help to provide a little more info regarding this in the beginning of the different sections in the results.

2.1 Answer: In response to this, we have revised the original manuscript very carefully according to the PLOS ONE format where the Materials and Methods were described after the Introduction section (Line 134). In Methods, we described where the dataset was obtained and how we defined it. 

Then, the methods used to compare sequences statistics of ITS region and its components among groups was described (Lines 205-295). We connected each subsequent section as follow:

Insertion-deletion polymorphic analysis of the ITS1 among Ampelomyces groups and between unrelated groups

We investigate the distribution of insertion-deletion polymorphisms in ITS1 sequences that could cause sequence length variations. Please see Lines 205-206.

ITS2 structure prediction

To evaluate the utility of ITS2 S2s, the ITS2 sequences... Please see Line 219.

Multiple ITS2 sequence-structure alignment and phylogenetic tree of Ampelomyces strains 

To determine Ampelomyces lineages based on its ITS2 S2s, a simultaneous multiple sequence alignment of ITS2 S2s was firstly estimated. Please see Lines 236 and 237.

A second phylogenetic tree based on ITS sequences was built to compare the distribution of Ampelomyces clades and not related fungi with those obtained using ITS2 S2s. Please see Lines 259 and 260.

For a comprehensible analysis, a second phylogenetic tree was constructed as described above, but the input data consisted of 26 ITS2 S2s… Please see Lines 268-269.

Evolutionary distance estimation among Ampelomyces lineages and putative Ampelomyces 

The resolution power of the phylogeny of Ampelomyces based on the ITS2 S2s to distinguish the ‘true’ Ampelomyces from putative Ampelomyces was evaluated by comparing the genetic divergences obtained among groups based on the simultaneous ITS2 sequences and structures alignment, and on the MSA of complete ITS sequences. Please see Lines 276-277.

Prediction of consensus ITS2 S2s from Ampelomyces 

For comparative purposes of sequence-structure motifs among fungal groups, … Please see Line 288.

3. The discussion concerning pseudogenes (lines 459-472) is vague and is some aspects problematic. The provided reference for pseudogenes (Harpke and Peterson) merely discuss this in the context of the 5.8S gene, not the ITS. This is understandable, as the term pseudogene is usually related with expression, and Harpke and Peterson indeed takes this into account by comparing 5.8S sequences from genomic DNA verses cDNA. It is however clear that the secondary structure of ITS is important for the splicing but whether the observed polymorphisms are enough to discuss loss-of-function paralogs is not clear to me. This part of the discussion needs to be made more specific. The last part of this section (line 466 onwards) is difficult to understand. Why would these pseudogenes be naturally selected? It would be more reasonable to assume release of selective constraints, leading to continuous accumulation of mutations, with the ultimate fate determined by genetic drift. Also, the evolution of the rRNA repeat is very much depending on concerted evolution, but this is not even mentioned nor discussed. In fact, as concerted evolution at least partially can be connected with meiosis and sexual reproduction, it would make sense to discuss this in the current manuscript.

Answer: The sequence and structure of the ITS region are essential for the formation of mature rRNA. The ITS2 contains specific cleavage sites and S2 motifs used by the enzymatic machinery responsible for processing pre-rRNA; and mutations that are adjacent to these critical regions have decreased or abolished 28S rRNA maturation (Côté et al. 2001). Our revision is in Lines 847-849. We have made these points clearer in the revised manuscript. 

We modified the entire paragraph for clarity in the Discussion section. We can assume the polymorphisms observed in the proximal 5.8S and 28S stem and the ITS2 can be either homogenized across the rRNA cistrons by unequal crossing over or gene conversion, and their fate determined by genetic drift. Nevertheless, the modifications observed in the proximal 5.8S and 28S stem may affect ITS2 processing in two ways. For example, the first variation of the hybridization model, an ITS2 derived from an Ampelomyces strain extracted from P. xanthii contains a short proximal 5.8S and 28S stem. This short stem may affect the enzymatic machinery that processes the ITS2 because the length of the stem is important for processing. Conversely, the second variation of the hybridization model, an ITS2 derived from an Ampelomyces isolated from U. necator, may have a more relaxed structure that is still functional. The function of the secondary molecules resulting from the processing of the ITS2 remains to be elucidated. In addition, it remains to be determined whether these secondary molecules provide a specific trait to the fungus. Indeed, it has recently been shown that ribosomes containing rRNA variants have altered gene expression and physiology (Parks et al., 2019). If the ITS sequence becomes a ghost pseudogene that favours the synthesis of proteins via ribosomal regulation, then it may be naturally selected by inhabiting a new environment under different biotic or abiotic pressures. Please see Lines 843-868.

4. The comparison of the secondary structure prediction and the phylogenetic analysis is lacking from the discussion. The authors repeatedly state that the secondary structure prediction of ITS is a valuable tool for identifying Ampelomyces sequences from the environment. This may indeed be the case, but it seems to me that the phylogenetic analysis provides exactly the same result and resolution. What exactly is the gain from using secondary structure prediction? I am lacking a critical discussion of this in the manuscript.

Answer: The discussion of the predicted ITS2 S2s were included in the discussion section. Please see Lines 805-817. 

The section of phylogenetic analysis based on ITS2 S2s was also discussed. Please see Lines 869-874.

Ampelomyces can be misidentified with other related picnidial mycoparasites that infect PMs. For initial identification purposes, cultures obtained by conidial isolation are grown and assayed for the formation of Ampelomyces pycnidia in PM conidiophores. Direct environmental DNA sequencing may facilitate identification, but the inter- and intra-specific distances of the ITS vary among some fungal groups. For instance, some ‘true’ Ampelomyces strains can have ITS sequence divergence up to 15% (Kiss and Nakasone, 1998 and Liang et al., 2007). 

This limitation may be resolved by the use of phylogenetic analysis based on ITS2 sequences and S2s. The different ITS2 models obtained in the whole mycoparasitic Ampelomyces population have clear differences to other fungi that have been misidentified as Ampelomyces. The genetic divergences between Ampelomyces and those fungi misidentified are higher when using the ITS2 S2 phylogenetic analysis than those from the analysis based on ITS sequences. Our revision is in Lines 869-874.

5. The analysis of length variation. First, it would be useful to provide a statement regarding the distribution of the ITS1 length variation between Ampelomyces and the other groups, is it due to specific indels or more evenly spread out over the sequence? 

Answer: We found that the number of indels between groups of Ampelomyces isolated from the same PM genera varied as assessed with DnaSP software. Visualization of a multiple sequence alignment of ITS1 sequences revealed that Ampelomyces ITS1 sequences extracted from PM of the genera Erysiphe, Golovinomyces and Podosphaera have indels evenly distributed along the sequence. In contrast, the ITS1 from Ampelomyces extracted from Podosphaera fusca in Korea, contained indels which were more distributed at the beginning of the ITS1 sequence (first 100 nucleotides), while the Ampelomyces derived from the same PM but sampled in China harboured indels at the end of the sequence. Conversely, Ampelomyces extracted from the genera Arthrocladiella, Oidium and the subgenus Pseudoidium had indels at the beginning and at the end of the sequence, while Ampelomyces derived from Phyllactinia contained indels localized at the end of the sequence. We also compared the number of indels for all Ampelomyces ITS1 sequences with those from putative Ampelomyces Groups 1 and 2 as well as the outgroup taxa. This analysis revealed that ITS1 sequences for all outgroup samples contained indels at various positions across the ITS1 sequence. Please see Lines 415-427.

5.1 Secondly, what was the basis for defining groups 1 and 2? This was not clear to me. 

Answer: Group 1 is defined as those Ampelomyces extracted from PMs or plants. Group 2 comprises those Ampelomyces extracted from non-plant material or PM-free environments. We clarified this point in the text (Please see Lines 156-159).

5.2 Third, why is the normalization of the length even necessary? The authors argue that it was necessary due to the use of different primers, but if all used sequences were complete full-length (ITS1-5.8S-ITS2) as stated, the primers should not matter. 

Answer: We agree with your comment. Some environmental ITS sequences deposited in the GenBank database do not contain the known motifs ATCATTA or TTGACC, which delimitate the full-length ITS. It makes difficult to delimitate the complete ITS sequence and delete the partial nucleotide sequences of the 18S and 28S. For instance, the sequence obtained from an air sample (Group 2), GenBank Identifier: LN808951, did not contain the motif TTGACC. In order to address your comment, we restructured the paragraph (Please see Lines 196-198). 

5.3. I assume that partial 18S and 28S sequences should be deleted before the analysis. This is not clear to me.

Answer: Yes, partial 18S and 28S rRNA sequences were deleted before the analysis. This has been clarified in the revised text. Please see line 195.

Minor issues:

1. Lines 41-43. The 5.8S gene is not assumed to evolve neutrally. Probably an error in the sentence.

Answer: The sentence has been corrected. The introduction has been modified for clarity. Please see Line 87.

2. Line 53. I would exchange “although only” with “and” in order to be a bit more positive concerning the application aspects. Commercialization is a complicated process, so it is great that two strains have gone the whole way to products.

Answer: The change has been made (Line 100).

3. Line 55. Delete “conidia, fruiting bodies”.

Answer: The change has been made (Line 102).

4. Line 97. Table 3 comes before Table 2.

Answer: We modified the order and content of the supplemental material as follows: Supplemental Material 1 consisted of the datasets used in each method while Supplemental material 2 contains DNA ITS and ITS2 secondary structures sequence analyses. Supplemental material 3 contains all the ITS2 sequence structures predicted by using the ITS2-DB. Please see Lines 898-902.

5. Line 109. Replace “extracted” with “retrieved”.

Answer: The change has been made. Please see Line 327.

6. Line 110. Replace “the mycoparasites” with “Ampelomyces mycoparasites”.

Answer: The change has been made (Lines 163 and 164).

7. The figures were of poor quality, but it may be related with the review versions.

Answer: The changes have been made. We used PACE to check the quality of the figures.

8. Line 274. “Major” should be “major”.

Answer: The change has been made. Line 525.

9. Line 282. “helices IIII” should be “helices III”, I guess.

Answer: Yes, it is helices III. The change has been made. Line 534.

10. Line 369. What is the significance of mentioning the free energy value for this structure but not for the others?

Answer: Despite we had a small number of samples, we noticed that the integrity of the hybridization model of the proximal stem of the ITS2 is compromised by energy values over -13.7. For instance, the variation observed in the ITS2 sequence of Ampelomyces from P. xanthii. Conversely, energy values between -15 and -19 conferred to the structure with some flexibility that may be necessary for post transcriptional processes of the pre-rRNA. We included a description of this finding in the Results section, Lines 451-453. 

On the other hand, we also included in the text (Please see Lines 637-639) the comparison of energy values for folding of the ITS2 S2 from putative Ampelomyces spp. and Ampelomyces spp. sensu stricto:

The ITS2 S2s from putative Ampelomyces spp. were homology modelled or directly folded and exhibited lower negative energy values (between -25.8 and -35.9) than those from Ampelomyces spp. sensu stricto (between -47.3 and -36.8) that were obtained by direct folding. This was another difference found between both fungal groups.

11. Line 410. Replace “from” with “with”.

Answer: The change has been made. Please see Line 706.

12. Figure 9 + legend. Bootstrap support values are typically reported as the percentage, e.g. 53 instead of 0.53.

Answer: The bootstrap support values were expressed as percentages. The change has been made in the figure together with its legend. Please see Line693.

13. Line 446. “neighbor-joining” should be “maximum likelihood”, right?

Answer: Yes, it is maximum likelihood. The change has been made. Line 668.

14. Lines 450-452. Is the Bartys et al reference really appropriate here? The reference seems to deal with hairpin structures in mRNA, while the importance in non-coding RNA may be different.

Answer: The reference was replaced (please see Line 670) with a suitable reference. Please see Line 810:

Svoboda, P. and Di Cara, A. (2006). Hairpin RNA: a secondary structure of primary importance. Cell. Mol. Life Sci. doi: 10.1007/s00018-005-5558-5.

References

Bruns, TD. (2001) ITS reality. Inoculum. 52:2–3.

Côté, C. and Peculis, B. (2001) Role of the ITS2-proximal stem and evidence for indirect recognition of processing sites in pre-rRNA processing in yeast. Nucleic Acids Res. 29:2106–2116.

Elder, JJ. and Turner, B. (1995) Concerted evolution of repetitive DNA sequences in eukaryotes. Q. Rev. Biol. 70:297–320.

Gazis, R., Rehner, S. and Chaverri, P. (2011) Species delimitation in fungal endophyte diversity studies and its implications in ecological and biogeographic inferences. Mol. Ecol. 20: 3001–3013.

Gomes, EA., Kasuya, MCM., de Barros, EG., Borges, AC. and Araujo, EF. (2002) Polymorphism in the internal transcribed spacer (ITS) of the ribosomal DNA of 26 isolates of ectomycorrhizal fungi. Genet. Mol. Biol. 25:477–483.

Gottschling, M. and Plötner, J. (2004) Secondary structure models of the nuclear internal transcribed spacer regions and 5.8S rRNA in Calciodinelloideae (Peridiniaceae) and other dinoflagellates. Nucleic Acids Res. 32:307–315.

Joseph, N., Krauskopf, E., Vera, MI. and Michot, B. (1999) Ribosomal internal transcribed spacer 2 (ITS2) exhibits a common core of secondary structure in vertebrates and yeast. Nucleic Acids Res. 27:4533–4540.

Keller, A., Schleicher, T., Schultz, J., Müller, T., Dandekar, T. and Wolf, M. (2009) 5.8S-28S rRNA interaction and HMM-based ITS2 annotation. Gene. 430:50−57.

Kiss, L., Pintye, A., Kovács, G, Jankovics, T., Fontaine, M., Harvey, N. et al. (2011) Temporal isolation explains host-related genetic differentiation in a group of widespread mycoparasitic fungi. Mol. Ecol. 20:1492−1507.

Kiss, L. and Nakasone, K. (1998) Ribosomal DNA internal transcribed spacer sequences do not support the species status of Ampelomyces quisqualis, a hyperparasite of powdery mildew fungi. Curr. Genet. 33:362–367.

Kovács, GM., Balázs, TK., Calonge, FD. and Martín, MP. (2011) The diversity of Terfezia desert truffles: New species and a highly variable species complex with intrasporocarpic nrDNA ITS heterogeneity. Mycologia 103:841–853.

Krüger, M., Krüger, C., Walker, C., Stockinger, H. and Schüssler, A. (2012) Phylogenetic reference data for systematics and phylotaxonomy of arbuscular mycorrhizal fungi from phylum to species level. New Phytol. 193:970–984.

Liang, C., Yang, J., Kovács, G., Szentiványi, O., Li, B., Xu, X., et al. (2007) Genetic diversity of Ampelomyces mycoparasites isolated from different powdery mildew species in China inferred from analyses of rDNA ITS sequences. Fungal Divers. 24:225−240.

Lindner, DL and Banik, MT. (2011) Intragenomic variation in the ITS rDNA region obscures phylogenetic relationships and inflates estimates of operational taxonomic units in genus Laetiporus. Mycologia 103:731–740.

Naidoo, K., Steenkamp, ET., Coetzee, MP., Wingfield, MJ. and Wingfield, BD. (2013) Concerted evolution in the ribosomal RNA cistron. PLoS One. 8:e59355.

Park, M., Choi, Y., Hong, S. and Shin, H. (2010) Genetic variability and mycohost association of Ampelomyces quisqualis inferred from phylogenetic analyses of ITS rDNA and actin gene sequences. Fungal Biol. 114:235−247.

Parks, MM., Kurylo, CM., Batchelder, JE., Vincent, CT. and Blanchard, SC. (2019) Implications of sequence variation on the evolution of rRNA. Chromosome Res. 27:89–93.

Schoch, CL., Seifert, KA., Huhndorf, S., Robert, V., Spouge, JL., Levesque, CA., Chen, W.; Fungal Barcoding Consortium; Fungal Barcoding Consortium Author List. (2012) Nuclear ribosomal internal transcribed spacer (ITS) region as a universal DNA barcode marker for Fungi. Proc. Natl.. Acad Sci U S A. 17: 6241-6246.

Simon, UK. and Weiss, M. (2008) Intragenomic variation of fungal ribosomal genes is higher than previously thought. Mol. Biol. Evol. 25:2251–2254.

Smith, ME., Douhan, GW. and Rizzo, DM. (2007) Intra-specific and intra-sporocarp ITS variation of ectomycorrhizal fungi as assessed by rDNA sequencing of sporocarps and pooled ectomycorrhizal roots from a Quercus woodland. Mycorrhiza 18:15–22.

Svoboda, P. and Di Cara, A. (2006) Hairpin RNA: a secondary structure of primary importance. Cell Mol. Life Sci. :901−908.

Zheng, D. and Gerstein, MB. (2007) The ambiguous boundary between genes and pseudogenes: the dead rise up, or do they? . Trends Genet. 23:219–224.

---

## [Decision Letter · Decision Letter 1]

24 May 2021

PONE-D-20-39142R1

The role of internal transcribed spacer 2 secondary structures in classifying mycoparasitic Ampelomyces

PLOS ONE

Dear Dr. Prahl,

Thank you for submitting your manuscript to PLOS ONE. After careful consideration, we feel that it has merit but does not fully meet PLOS ONE’s publication criteria as it currently stands. Therefore, we invite you to submit a revised version of the manuscript that addresses the points raised during the review process.

We look forward to receiving your revised manuscript.

Kind regards,

Kandasamy Ulaganathan

Academic Editor

PLOS ONE

Journal Requirements:

Reviewers' comments:

Reviewer's Responses to Questions

**Comments to the Author**

1. If the authors have adequately addressed your comments raised in a previous round of review and you feel that this manuscript is now acceptable for publication, you may indicate that here to bypass the “Comments to the Author” section, enter your conflict of interest statement in the “Confidential to Editor” section, and submit your "Accept" recommendation.

Reviewer #1: All comments have been addressed

Reviewer #2: All comments have been addressed

2. Is the manuscript technically sound, and do the data support the conclusions?

Reviewer #1: Yes

Reviewer #2: Yes

3. Has the statistical analysis been performed appropriately and rigorously? 

Reviewer #1: Yes

Reviewer #2: N/A

4. Have the authors made all data underlying the findings in their manuscript fully available?

Reviewer #1: Yes

Reviewer #2: Yes

5. Is the manuscript presented in an intelligible fashion and written in standard English?

Reviewer #1: Yes

Reviewer #2: Yes

6. Review Comments to the Author

Reviewer #1: The authors have answered all my concerns in an adequate manner. I have no further questions or comments.

Reviewer #2: This manuscript "The role of internal transcribed spacer 2 secondary structures in classifying mycoparasitic Ampelomyces" is well written and suitable to be published in Plos One. I made few comments in the manuscript for your attention.

7. PLOS authors have the option to publish the peer review history of their article (what does this mean?). If published, this will include your full peer review and any attached files.

Reviewer #1: No

Reviewer #2: No

---

## [Author Response · Author response to Decision Letter 1]

3 Jun 2021

All amendments have been made. Thank you.

---

## [Editor Report · Decision Letter 2]

14 Jun 2021

The role of internal transcribed spacer 2 secondary structures in classifying mycoparasitic Ampelomyces

PONE-D-20-39142R2

Dear Dr. Prahl,

We’re pleased to inform you that your manuscript has been judged scientifically suitable for publication and will be formally accepted for publication once it meets all outstanding technical requirements.

Kind regards,

Kandasamy Ulaganathan

Academic Editor

PLOS ONE
---

## [Editor Report · Acceptance letter]

21 Jun 2021

PONE-D-20-39142R2 

The role of internal transcribed spacer 2 secondary structures in classifying mycoparasitic *Ampelomyces*

Dear Dr. Prahl:

I'm pleased to inform you that your manuscript has been deemed suitable for publication in PLOS ONE. Congratulations! Your manuscript is now with our production department. 

Kind regards, 

on behalf of

Dr. Kandasamy Ulaganathan 

Academic Editor

PLOS ONE